# Environmental cystine drives glutamine anaplerosis and sensitizes cancer cells to glutaminase inhibition

Alexander Muir[1], Laura V Danai[1], Dan Y Gui[1], Chiara Y Waingarten[1], Caroline A Lewis[2], Matthew G Vander Heiden[1,3]*

[1]Koch Institute for Integrative Cancer Research and Department of Biology, Massachusetts Institute of Technology, Cambridge, United States; [2]Whitehead Institute for Biomedical Research, Cambridge, United States; [3]Dana-Farber Cancer Institute, Boston, United States

**Abstract** Many mammalian cancer cell lines depend on glutamine as a major tri-carboxylic acid (TCA) cycle anaplerotic substrate to support proliferation. However, some cell lines that depend on glutamine anaplerosis in culture rely less on glutamine catabolism to proliferate in vivo. We sought to understand the environmental differences that cause differential dependence on glutamine for anaplerosis. We find that cells cultured in adult bovine serum, which better reflects nutrients available to cells in vivo, exhibit decreased glutamine catabolism and reduced reliance on glutamine anaplerosis compared to cells cultured in standard tissue culture conditions. We find that levels of a single nutrient, cystine, accounts for the differential dependence on glutamine in these different environmental contexts. Further, we show that cystine levels dictate glutamine dependence via the cystine/glutamate antiporter xCT/*SLC7A11*. Thus, xCT/*SLC7A11* expression, in conjunction with environmental cystine, is necessary and sufficient to increase glutamine catabolism, defining important determinants of glutamine anaplerosis and glutaminase dependence in cancer.
DOI: https://doi.org/10.7554/eLife.27713.001

*For correspondence:
mvh@mit.edu

## Introduction

Altered metabolism is a hallmark of cancer (*Hanahan and Weinberg, 2011*) and reflects the increased energetic and biosynthetic demands of proliferating cancer cells (*DeBerardinis and Chandel, 2016*; *Vander Heiden and DeBerardinis, 2017*). Indeed, one of the earliest noted biochemical differences between cancerous and normal tissues is increased glucose uptake and glycolysis in tumors (*Cori and Cori, 1925*; *Warburg et al., 1927*). Beyond increased glucose metabolism in tumors, another long standing observation is that many cancer cells consume substantial amounts of glutamine in culture, far in excess of other amino acids (*Eagle, 1955*; *Jain et al., 2012*). Consistent with this, many cancer cell lines depend on extracellular glutamine for proliferation in vitro, even though glutamine is nominally non-essential and can be synthesized from other nutrients. This contributed to the concept that some cancer cells are glutamine addicted (*Altman et al., 2016*; *Wise and Thompson, 2010*).

Glutamine can be used to support proliferation in multiple ways. It is a proteinogenic amino acid, can act as a nitrogen donor for the synthesis of amino acids as well as nucleotides, and glutamine can contribute carbon to the TCA cycle to replace cycle intermediates that are removed for the production of biomass, a process termed anaplerosis (*Altman et al., 2016*; *Daye and Wellen, 2012*; *DeBerardinis and Cheng, 2010*; *Hensley et al., 2013*). Analysis of the fate of glutamine nitrogen in proliferating cancer cells in vitro suggests that most glutamine nitrogen is excreted as ammonia and alanine, suggesting that the high rate of glutamine consumption is not due to nitrogen demand

**eLife digest** Cancer cells need to consume certain nutrients in order to grow, and some cancer drugs work by affecting the ability of the cells to use these nutrients. For decades researchers have grown cancer cells in petri dishes with standard nutrient formulations (also known as tissue culture), but the nutrients available to cancer cells in tissue culture are not the same as those found in the body.

Cancer cells growing in tissue culture consume large amounts of a nutrient called glutamine. These cells die when exposed to a class of drugs called glutaminase inhibitors that prevent them from processing glutamine. However, when these same cancer cells grow as tumors in animals, they process less glutamine and are not affected by glutaminase inhibitors. So what differences are there between growing cancer cells in tumors and tissue culture that explain this different reliance on glutamine?

Muir et al. reasoned that changing the levels of nutrients available to cancer cells might change what these cells consume, and so grew human cancer cells in cow serum (which has a similar nutrient content to blood in humans and other mammals). Indeed, these cells consumed less glutamine and responded to glutaminase inhibitors in a way that is similar to how tumors in the body respond to these drugs. Comparing the nutrient content of cow serum and typical tissue culture formulations revealed that high levels of the nutrient cystine cause the cells to rely more on glutamine.

The results presented by Muir et al. suggest that cancer cells in tumors could be made to consume more glutamine and that this would make them sensitive to glutaminase inhibitors – a possibility that will be studied in future work. Exposing cultured cancer cells to nutrient levels closer to those found in the body may also better predict how tumor cells use nutrients and respond to some treatments.

DOI: https://doi.org/10.7554/eLife.27713.002

(*DeBerardinis et al., 2007*). Instead, glutamine metabolism is important for TCA cycle anaplerosis in multiple contexts (*Altman et al., 2016*; *DeBerardinis et al., 2007*; *Yuneva et al., 2007*). Across cancer cell lines, the majority of aspartate, glutamate and TCA cycle metabolites are glutamine-derived (*Altman et al., 2016*; *Wise and Thompson, 2010*), with production of amino acids being a major fate of anaplerotic glutamine carbon (*Hosios et al., 2016*). In line with these results, proliferation of many cell types following glutamine starvation is rescued by providing an alternative source of the TCA cycle intermediates α-ketoglutarate (αKG) (*van den Heuvel et al., 2012*; *Wise et al., 2008*; *Wise et al., 2011*) or oxaloacetate (*Patel et al., 2016*). Collectively, these results suggest that anaplerosis contributes to the large cellular consumption of glutamine in culture, and to dependence on this amino acid for cell proliferation and survival.

Glutamine can enter the TCA cycle through multiple metabolic routes. First, several transporters are capable of transporting glutamine into cells (*Hediger et al., 2013*; *Hyde et al., 2003*). Relevant to cancer, the neutral amino acid transporters ASCT2/*SLC1A5* and LAT1/*SLC7A5* are known to have higher expression in certain tumors, and can mediate glutamine uptake in cell lines derived from these tumors (*Bhutia et al., 2015*; *Pochini et al., 2014*). Intracellularly, glutamine is converted to glutamate either by donating the amide nitrogen for the production of nucleotides or asparagine, or by glutaminase activity (encoded by *GLS1 or GLS2*), which produces glutamate and ammonia from glutamine (*Curthoys and Watford, 1995*; *Krebs, 1935*). In proliferating cells, glutaminase activity can be the primary driver of glutamate production from glutamine, as amide nitrogen incorporation into mass is low compared to release of glutamine nitrogen as ammonia (*Brand, 1985*; *Brand et al., 1986*; *Hosios et al., 2016*; *Wise et al., 2008*). Genetic or pharmacological loss of *GLS1* activity depletes TCA metabolites and slows proliferation of a variety of cancer cell lines in culture (*Cheng et al., 2011*; *Gameiro et al., 2013*; *Gao et al., 2009*; *Gross et al., 2014*; *Le et al., 2012*; *Seltzer et al., 2010*; *Son et al., 2013*; *Timmerman et al., 2013*; *van den Heuvel et al., 2012*; *Wang et al., 2010*; *Yuneva et al., 2012*). This has led to interest in targeting glutaminase activity therapeutically, and the glutaminase inhibitor CB-839 is being evaluated in clinical trials to treat cancer (*Gross et al., 2014*). In the last step of glutamine carbon entry into the TCA cycle, glutamate produced from glutamine is converted to αKG by either transamination reactions or by glutamate

dehydrogenase to produce αKG as an anaplerotic TCA cycle intermediate (*Moreadith and Lehninger, 1984*). Rapidly proliferating cells have been shown to preferentially use transamination reactions for αKG production, consistent with their increased need for nitrogen for biosynthetic demand (*Coloff et al., 2016*). Finally, consistent with these observations of increased glutamine catabolism and dependence in rapidly proliferating cultured cells, glutamine catabolic pathways are controlled by oncogene expression and upregulated in many cancer cell lines (*Altman et al., 2016*).

Tumor cell environment can also influence dependence on glutaminase for anaplerosis and proliferation. Tracing of glucose and glutamine fate in tumors derived from human non-small cell lung cancer (NSCLC) and mouse *KRAS*-driven NSCLC found that these tumors rely more on glucose than on glutamine for TCA cycle anaplerosis (*Davidson et al., 2016*; *Hensley et al., 2016*; *Sellers et al., 2015*). This correlated with resistance of these tumors to glutaminase inhibition in vivo; but surprisingly, cell lines derived from these tumors with decreased glutamine catabolism showed dramatically increased glutamine anaplerosis and sensitivity to glutaminase inhibition when cultured in vitro (*Davidson et al., 2016*). These results suggest that, even for cells capable of glutamine catabolism, environmental differences between tumors in vivo and tissue culture dictates the use of glutamine for anaplerosis.

We describe the identification of an environmental factor that contributes to differential glutamine anaplerosis between human NSCLC tumors growing in vivo and cell lines cultured in vitro. Nutrient availability is different for cancer cells growing as tumors in vivo compared to culture conditions in vitro, and we explored the role of nutrient environment by culturing the human A549 NSCLC cell line in adult bovine serum, a medium that more closely models in vivo nutrient levels. In this condition, the contribution of glutamine carbon to the TCA cycle decreases to levels observed for A549 tumors growing in vivo. Culture in adult bovine serum also induces glutaminase inhibitor resistance, a phenotype seen for NSCLC tumors in vivo. Deconvolution of differences in media composition showed that levels of a single nutrient, cystine (the oxidized dimer of the amino acid cysteine), largely dictates the observed differences in glutamine anaplerosis and dependence. Further, we find that cystine regulation of glutamine anaplerosis depends on expression of the cystine/glutamate antiporter xCT (encoded by the gene *SLC7A11*). Lastly, we find that administration of cystine to tumor bearing mice increases glutamine use by tumors in vivo. Collectively, these results suggest that environmental cystine availability and xCT/*SLC7A11* expression are critical determinants of glutamine anaplerosis and glutaminase dependence. They also highlight how nutrient conditions can impact cell metabolism.

## Results

### Cells in vivo or cultured in adult bovine serum exhibit limited glutamine catabolism compared to cells cultured in standard tissue culture conditions

Mutant *Kras*-driven mouse lung cancer cells exhibit differences in glutamine metabolism dictated by their environment, such that glutamine extensively labels TCA cycle intermediates in cell culture but not in tumors (*Davidson et al., 2016*). To confirm this finding, we examined glutamine metabolism in mutant *KRAS*-driven human A549 lung cancer cells cultured in multiple environments. For these studies, we used gas chromatography-mass spectrometry (GC-MS) to trace the fate of glutamine where all five carbons are $^{13}C$ labeled ($[U-^{13}C_5]$glutamine) in each environment, focusing on whether glutamine carbon contributed to the TCA cycle (*Figure 1A*). First, we verified that glutamine was not a major source of TCA cycle carbon in subcutaneous A549 xenograft tumors. For these studies $[U-^{13}C_5]$glutamine was infused into tumor bearing animals for 6 hr to achieve a final enrichment of ~35% labeled glutamine in plasma and ~25% in tumors (*Figure 1B*). Consistent with previous results (*Davidson et al., 2016*), there was little labeling of intratumoral glutamate, TCA cycle intermediates, and aspartate from $[U-^{13}C_5]$glutamine despite the presence of labeled glutamine in the tumors (*Figure 1B*). These findings suggest that glutamine is not a major source of TCA cycle carbon in A549-derived tumors in vivo. Normalization of tumor m + 5 glutamate label to m + 5 glutamine labeling indicates that only ~25% of glutamate is derived from glutamine in these tumors. In contrast, when A549 cells are cultured in RPMI-based media with $[U-^{13}C_5]$glutamine added to a similar enrichment of ~33%, we find that ~27% of glutamate is glutamine derived, which when normalized

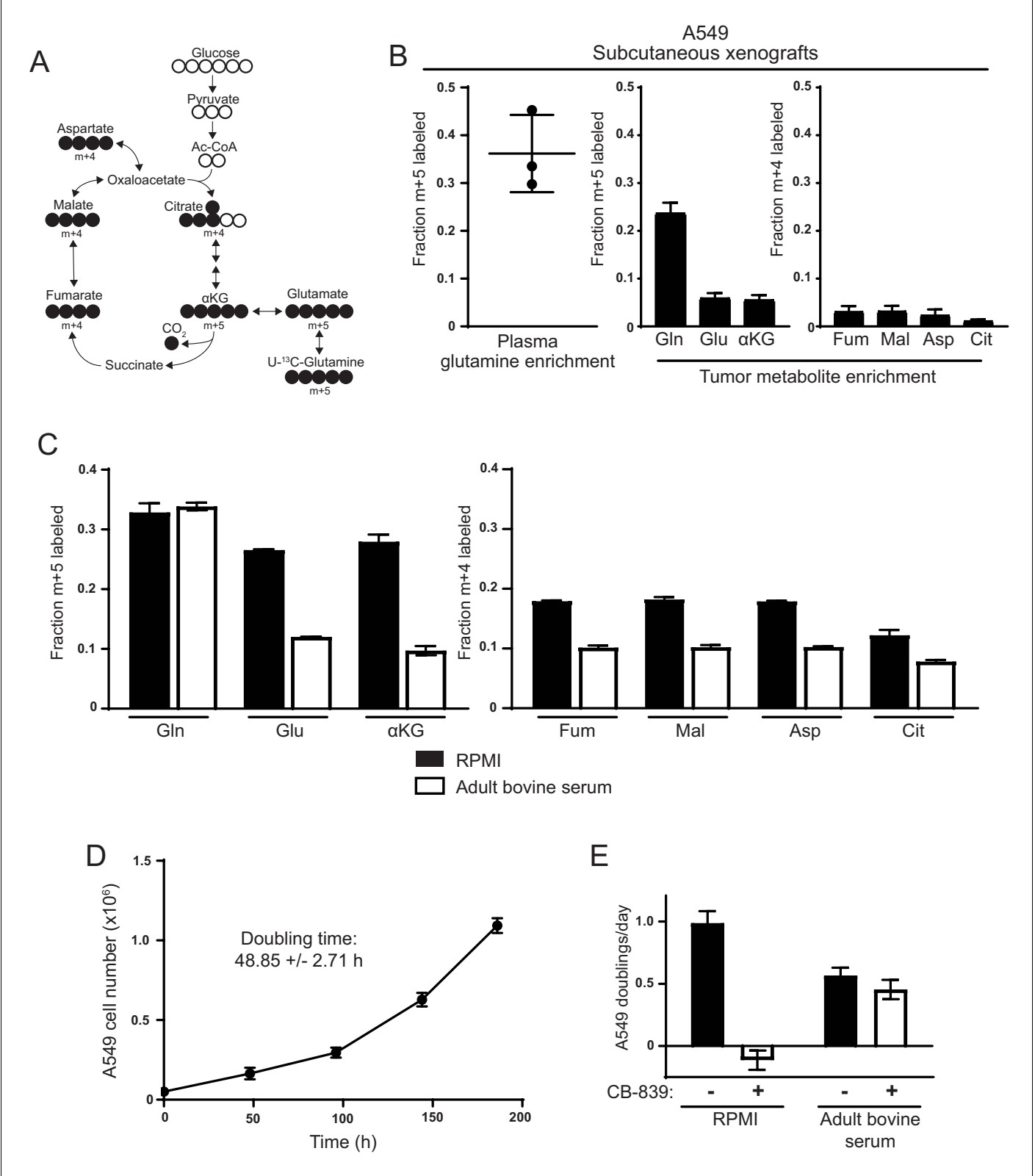

**Figure 1.** Decreased use of glutamine for TCA cycle anaplerosis by A549 cells in tumors, and when cultured in adult bovine serum, compared to RPMI. (**A**) Diagram detailing how uniformly-labeled glutamine ([U-$^{13}$C$_5$]glutamine) can be metabolized to generate labeled glutamate, aspartate and TCA cycle intermediates via oxidative metabolism. (**B**) *Left* Plasma fractional labeling of fully labeled glutamine (m + 5) in A549 tumor bearing mice following a 6 hr infusion of [U-$^{13}$C$_5$]glutamine (n = 3). *Right* Intratumoral fractional labeling of glutamine (m + 5), glutamate (m + 5), α-ketoglutarate (m + 5),

*Figure 1 continued on next page*

*Figure 1 continued*

fumarate (m + 4), malate (m + 4), aspartate (m + 4) and citrate (m + 4) following a 6 hr infusion of [U-$^{13}$C$_5$]glutamine (n = 3). (**C**) M + 5 fractional labeling of glutamine, glutamate and α-ketoglutarate, and m + 4 fractional labeling of fumarate, malate, aspartate and citrate for A549 cells cultured for 8 hr in RPMI or adult bovine serum with [U-$^{13}$C$_5$]glutamine added to ~33% enrichment (n = 3). (**D**) A549 cell counts over time when cultured continuously in adult bovine serum for eight days (n = 3, each time point). Doubling time was determined by non-linear regression of an exponential growth equation to the growth curve. (**E**) Proliferation rate of A549 cells cultured in RPMI or adult bovine serum with vehicle (DMSO) or 1 μM CB-839 (n = 3) as indicated. For all panels, the values represent the mean and the error bars represent ± SD.

DOI: https://doi.org/10.7554/eLife.27713.003

The following source data and figure supplements are available for figure 1:

**Source data 1.** Mass isotopomer distributions for all metabolites analyzed by GC-MS in *Figure 1*.
DOI: https://doi.org/10.7554/eLife.27713.007
**Source data 2.** Mass isotopomer distributions for all metabolites analyzed by GC-MS in *Figure 1—figure supplement 1*.
DOI: https://doi.org/10.7554/eLife.27713.008
**Source data 3.** Mass isotopomer distributions for all metabolites analyzed by GC-MS in *Figure 1—figure supplement 2*.
DOI: https://doi.org/10.7554/eLife.27713.009
**Source data 4.** Mass isotopomer distributions for all metabolites analyzed by GC-MS in *Figure 1—figure supplement 3*.
DOI: https://doi.org/10.7554/eLife.27713.010
**Figure supplement 1.** Glutamine labeling of metabolites in A549 cells cultured in adult bovine serum reaches isotopic steady state by 8 hr.
DOI: https://doi.org/10.7554/eLife.27713.004
**Figure supplement 2.** Decreased glutamine anaplerosis is not unique to serum or bovine derived blood products.
DOI: https://doi.org/10.7554/eLife.27713.005
**Figure supplement 3.** Decreased glutamine anaplerosis in adult bovine serum is not unique to A549 cells.
DOI: https://doi.org/10.7554/eLife.27713.006

to glutamine enrichment indicates that 80% of glutamate is derived from glutamine under these conditions (*Figure 1C*). Additionally, when the cells are cultured in RPMI, more than 60% of the carbon in aspartate and other TCA cycle intermediates was derived from glutamine (*Figure 1C*). This is consistent with glutamine being a major TCA cycle carbon source for A549 cells in culture, and demonstrates that compared to standard culture conditions tumors derived from these cells use less glutamine in vivo.

To begin to examine the environmental factors that contribute to this difference in glutamine metabolism, we cultured cells in 100% adult bovine serum with no additional nutrients based on the notion that this may better reflect the nutrient levels available from the circulation to the cells in tumors. A549 cells grow exponentially with a doubling time of ~48 hr when cultured in bovine serum (*Figure 1D*) and can be maintained indefinitely in such medium (see Materials and methods for details). We added [U-$^{13}$C$_5$]glutamine to adult bovine serum to achieve an enrichment of ~33% that is comparable to the labeled glutamine enrichment observed in plasma and tumors in vivo. We then traced the fate of $^{13}$C carbon into the TCA cycle and found substantially less labeling of glutamate, aspartate and TCA cycle metabolites compared to cells cultured in RPMI-based media (*Figure 1C*). In fact, only ~35% of glutamate was estimated to be derived from glutamine when cells were cultured in adult bovine serum. We verified that glutamine labeling reached isotopic steady state after 8 hr of culture in adult bovine serum (*Figure 1—figure supplement 1*), suggesting that these labeling differences were not explained by differences in labeling kinetics. Importantly, the decrease in glutamine-derived label observed in A549 cells cultured in adult bovine serum was not specific to bovine serum or to serum products, as the contribution of glutamine carbon to glutamate and TCA cycle intermediates was similarly reduced in adult bovine heparinized plasma and adult human serum (*Figure 1—figure supplement 2*). Reduced TCA cycle labeling from glutamine when cells are cultured in adult bovine serum is not limited to A549 cells. [U-$^{13}$C$_5$]glutamine labeling of TCA cycle intermediates was higher in cell lines derived from cancers arising in diverse tissues when cultured in RPMI compared to culture in adult bovine serum (*Figure 1—figure supplement 3*). Thus, culturing cancer cells in serum or plasma results in reduced glutamine catabolism to support TCA cycle anaplerosis.

Sensitivity of *Kras*-driven mouse lung cancer cells to the glutaminase inhibitor CB-839 correlates with glutamine carbon contribution to the TCA cycle, such that tumors derived from these cells in vivo were insensitive to CB-839 while proliferation of the same cells is inhibited in vitro

(*Davidson et al., 2016*). Consistent with these findings, we find that A549 cells cultured in RPMI are also sensitive to CB-839, but the same cells become resistant to CB-839 when cultured in adult bovine serum (*Figure 1E*). Thus, A549 cells cultured in bovine serum exhibit decreased glutamine metabolism and are resistant to glutaminase inhibitors, adopting a metabolic phenotype with regards to glutamine metabolism that is more similar to lung tumors in vivo than to lung cancer cells cultured in standard tissue culture media.

## Differences in the small molecule fraction between RPMI and adult bovine serum affect glutamine use for TCA cycle anaplerosis

We next sought to identify factors that are different between RPMI and adult bovine serum that account for the observed differences in glutamine anaplerosis and dependence. Dialysis experiments were used to determine whether differences in the small molecule (<3.5 kDa) or large molecule (>3.5 kDa) fractions accounted for differences in glutamine contribution to the TCA cycle. First, we dialyzed a small volume of RPMI against large volumes of adult bovine serum using 3.5 kDa cutoff dialysis cassettes (*Figure 2A*). This medium termed 'adult bovine serum → RPMI,' contained the small molecule fraction from adult bovine serum and the large molecule fraction from RPMI-based media. Similarly, we dialyzed adult bovine serum against RPMI to generate 'RPMI → adult bovine serum,' containing RPMI small molecules and the adult bovine serum large molecule fraction (*Figure 2A*). Cells could be cultured in both media conditions, allowing us to trace the fate of [U-$^{13}$C$_5$]glutamine in A549 cells cultured in each media. Similar to cells grown in adult bovine serum, cells grown in 'adult bovine serum → RPMI' exhibited low fractional labeling of glutamate, aspartate and TCA cycle metabolites (*Figure 2B*). In contrast, cells cultured in 'RPMI → adult bovine serum' exhibited higher labeling of aspartate and TCA cycle intermediates from glutamine, similar to cells cultured in RPMI (*Figure 2B*). These differences in glutamine contribution to the TCA cycle also correlated with the ability of CB-839 to inhibit proliferation, as A549 cell proliferation is inhibited by CB-839 when cells are cultured in 'RPMI → adult bovine serum' but is largely unaffected when cultured in 'adult bovine serum → RPMI' (*Figure 2C*). Similar dialysis experiments were performed with DMEM and adult bovine serum. Consistent with the RPMI dialysis experiments, 'adult bovine serum → DMEM' cultured cells displayed levels of glutamine anaplerosis and CB-839 sensitivity that were similar to cells cultured in adult bovine serum, while 'DMEM → adult bovine serum' had increased CB-839 sensitivity, similar to cells cultured in DMEM (*Figure 2—figure supplement 1*). Taken together, these experiments suggest that differences in the small molecule fraction of standard tissue culture media and adult bovine serum largely explain the differences in glutamine anaplerosis and glutaminase inhibitor sensitivity.

## Environmental cystine availability increases glutamine anaplerosis

We next considered differences in the small molecule fraction of adult bovine serum and RPMI/ DMEM that could give rise to the differences in glutamine anaplerosis and CB-839 sensitivity. RPMI and DMEM both contain excess glucose, amino acids and micronutrients compared to blood (*Mayers and Vander Heiden, 2015*). In fact, re-formulation of DMEM to have more physiological concentrations of glucose, amino acids and other nutrients were previously reported to alter central carbon metabolism and glutamine dependence (*Schug et al., 2015*; *Tardito et al., 2015*). In order to determine if excess nutrients in standard media formulations potentiate glutamine anaplerosis and dependence, we determined the concentration of amino acids, glucose, and pyruvate in the sera used to culture cells in this study (*Table 1*). We supplemented the adult bovine serum with amino acids, glucose, and pyruvate to match levels found in RPMI or DMEM. Levels of vitamins and micronutrients were added to adult bovine serum at the RPMI or DMEM concentration. Addition of RPMI nutrients (glucose, amino acids, vitamins and micronutrients) to adult bovine serum resulted in increased glutamine contribution to the TCA cycle in A549 cells (*Figure 3A*). Addition of nutrients to adult bovine serum to levels found in either RPMI or DMEM also caused increased A549 sensitivity to CB-839 (*Figure 3B*, *Figure 3—figure supplement 1A*). Thus, culturing cells in nutrient levels found in standard tissue culture media is sufficient to increase glutamine catabolism and dependence in these cells.

We next added different subsets of RPMI or DMEM nutrients to adult bovine serum to identify which nutrient(s) promote increased glutamine utilization and dependence. Addition of amino acids

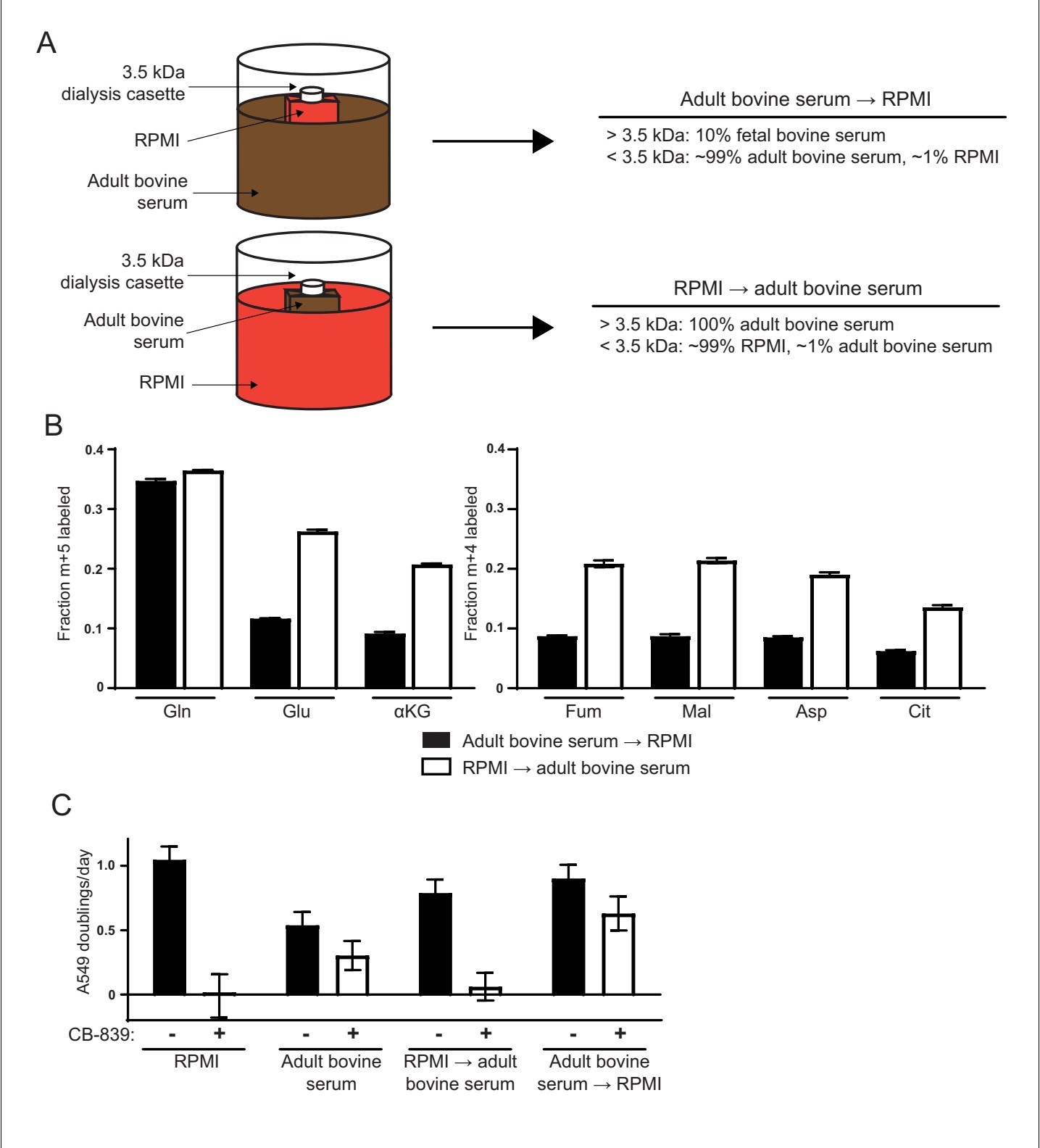

**Figure 2.** Differences in the small molecule (<3.5 kDa) fraction between RPMI and adult bovine serum account for differences in glutamine anaplerosis and sensitivity to glutaminase inhibition. (**A**) Diagram detailing the generation of *top* 'Adult bovine serum → RPMI' and *bottom* 'RPMI → adult bovine serum'. To make each dialyzed medium, 210 mL of RPMI or adult bovine serum was dialyzed twice against 4 L of adult bovine serum or RPMI respectively using 70 mL capacity 3.5 kDa cutoff dialysis cassettes. (**B**) M + 5 fractional labeling of glutamine, glutamate and α-ketoglutarate, and m + 4

*Figure 2 continued on next page*

*Figure 2 continued*

labeling of fumarate, malate, aspartate and citrate for A549 cells cultured for 8 hr in 'Adult bovine serum → RPMI' and 'RPMI → adult bovine serum' with [U-$^{13}$C$_5$]glutamine added to each media at ~33% enrichment (n = 3). (C) Proliferation rates of A549 cells cultured in RPMI, adult bovine serum, 'RPMI → adult bovine serum', 'Adult bovine serum → RPMI' with vehicle (DMSO) or 1 µM CB-839 (n = 3). For all panels, the values represent the mean and the error bars represent ± SD.

DOI: https://doi.org/10.7554/eLife.27713.011

The following source data and figure supplement are available for figure 2:

**Source data 1.** Mass isotopomer distributions for all metabolites analyzed by GC-MS in *Figure 2*.

DOI: https://doi.org/10.7554/eLife.27713.013

**Source data 2.** Mass isotopomer distributions for all metabolites analyzed by GC-MS in *Figure 2—figure supplement 1*.

DOI: https://doi.org/10.7554/eLife.27713.014

**Figure supplement 1.** Differences in the small molecule (<3.5 kDa) fraction between DMEM and adult bovine serum account for differences in glutamine anaplerosis and sensitivity to glutaminase inhibition.

DOI: https://doi.org/10.7554/eLife.27713.012

**Table 1.** Amino acid, glucose, pyruvate and lactate concentrations in all media used in this study compared to human plasma clinical reference values

| Metabolite | Human male plasma reference range* [µM] | RPMI-1640 with 10% dialyzed fetal bovine serum [µM] | DMEM with 10% dialyzed fetal bovine serum [µM] | Adult bovine serum† [µM] | Adult bovine heparinized plasma† [µM] | Adult human serum† [µM] |
|---|---|---|---|---|---|---|
| Alanine | 146–494 | 0 | 0 | 314 ± 6 | 321 ± 3 | 670 ± 13 |
| Arginine | 28–96 | 1034 | 360 | 312 ± 10 | 150 ± 9 | 216 ± 8 |
| Aspargine | 32–92 | 341 | 0 | 17 ± 1 | 19 ± 1 | 81 ± 2 |
| Aspartate | 2–9 | 135 | 0 | 7.4 ± 0.4 | 5.3 ± 0.4 | 59.2 ± 1.3 |
| Cystine | 24–54 | 187 | 180 | 0.3 ± 0.1 | 2 ± 0.1 | 3.4 ± 0.2 |
| Glutamate | 6–62 | 122 | 0 | 192 ± 3 | 120 ± 1 | 348 ± 5 |
| Glutamine | 466–798 | 1849 | 3600 | 183 ± 4 | 291 ± 1 | 409 ± 7 |
| Glycine | 147–299 | 120 | 360 | 302 ± 6 | 221 ± 1 | 409 ± 8 |
| Histidine | 72–108 | 87 | 180 | 8.9 ± 0.2 | 8.2 ± 0.2 | 17.7 ± 0.3 |
| Isoleucine | 46–90 | 344 | 720 | 112 ± 2 | 84 ± 1 | 114 ± 1 |
| Leucine | 113–205 | 344 | 720 | 218 ± 3 | 172 ± 1 | 231 ± 5 |
| Lysine | 135–243 | 197 | 720 | 92 ± 2 | 146 ± 1 | 206 ± 4 |
| Methionine | 13–37 | 91 | 180 | 21 ± 1 | 26 ± 1 | 39 ± 1 |
| Phenylalanine | 46–74 | 82 | 360 | 92 ± 2 | 68 ± 1 | 144 ± 3 |
| Proline | 97–297 | 157 | 0 | 88 ± 2 | 61 ± 1 | 301 ± 4 |
| Serine | 89–165 | 257 | 360 | 119 ± 2 | 74 ± 1 | 237 ± 5 |
| Threonine | 92–180 | 151 | 720 | 63 ± 1 | 50 ± 1 | 192 ± 3 |
| Tryptophan | 25–65 | 22 | 72 | ND | ND | ND |
| Tyrosine | 37–77 | 99 | 360 | 56 ± 1 | 47 ± 1 | 101 ± 3 |
| Valine | 179–335 | 154 | 720 | 251 ± 5 | 181 ± 2 | 325 ± 6 |
| Glucose | | 9990 | 22500 | 3932 ± 26 | 8403 ± 33 | 2407 ± 23 |
| Pyruvate | 27–160 | 0 | 900 | 9.4 ± 0.2 | 70.8 ± 0.5 | 10.3 ± 0.7 |
| Lactate | | 0 | 0 | 10878 ± 217 | 9291 ± 61 | 9250 ± 178 |

* Shown is the range ±2 standard deviations from the mean value for the indicated metabolite. These values are from (*Blau, 2003*).

† Shown are the mean values ± the standard error of the mean for the indicated metabolites as determined by GC-MS (see Materials and methods for detailed procedure), except glucose concentration, which was determined using a YSI bioanalyzer. All samples were analyzed in triplicate. ND indicates that the metabolite was not detected.

DOI: https://doi.org/10.7554/eLife.27713.015

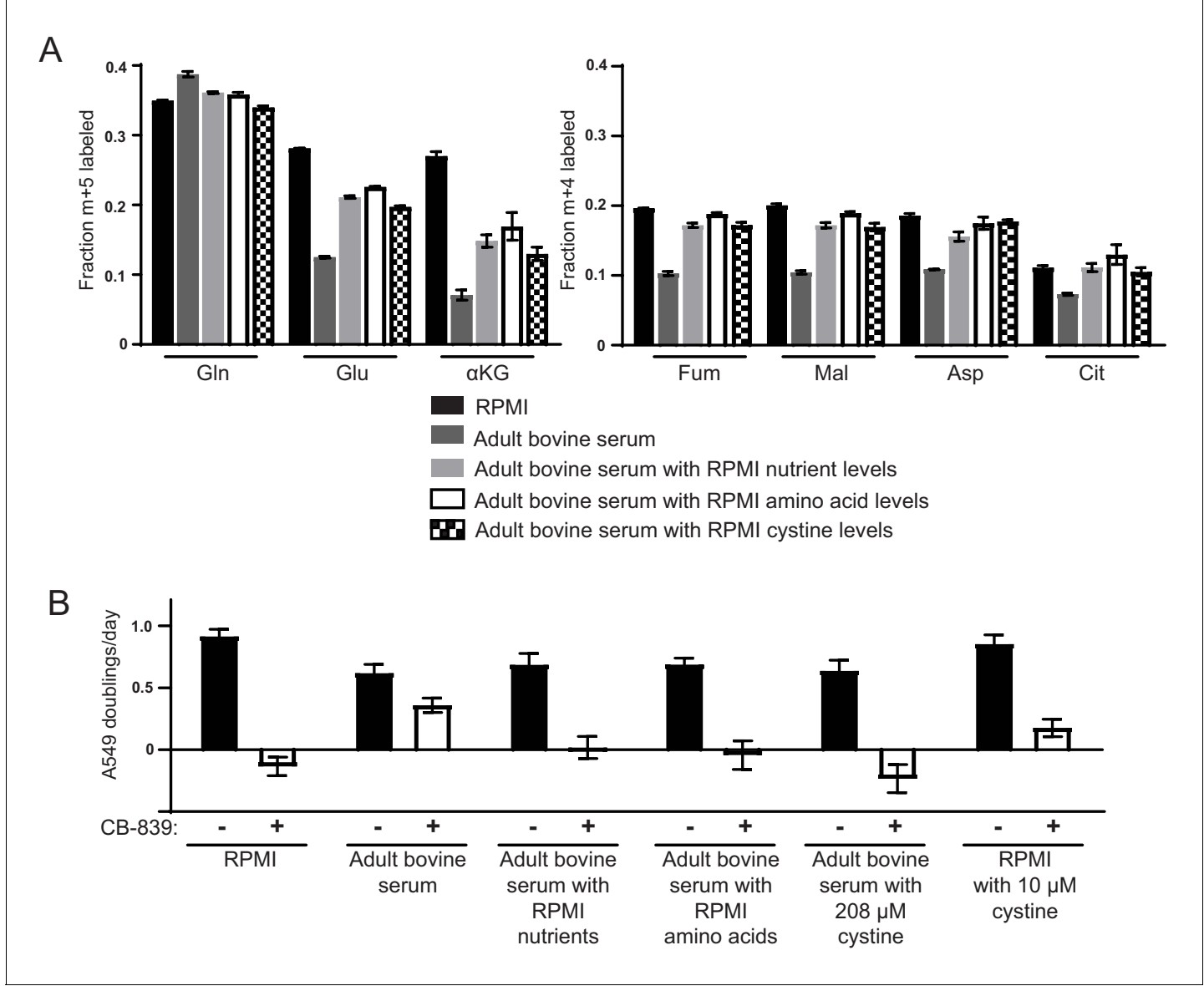

**Figure 3.** High levels of cystine enhance glutamine anaplerosis and potentiate the effects of the glutaminase inhibitor CB-839. (**A**) M + 5 fractional labeling of glutamine, glutamate and α-ketoglutarate, and m + 4 labeling of fumarate, malate, aspartate and citrate is shown for A549 cells cultured for 8 hr in RPMI, adult bovine serum, adult bovine serum with RPMI nutrient levels, adult bovine serum with RPMI amino acid levels and adult bovine serum with 208 μM cystine added (RPMI cystine levels). Each medium included [U-$^{13}$C$_5$]glutamine added to ~33% enrichment (n = 3). (**B**) The proliferation of A549 cells cultured in the same media defined in (**A**) and RPMI containing 10 μM cystine with vehicle (DMSO) or 1 μM CB-839 as indicated (n = 3). For all panels, the values represent the mean and the error bars represent ± SD.

DOI: https://doi.org/10.7554/eLife.27713.016

The following source data and figure supplements are available for figure 3:

**Source data 1.** Mass isotopomer distributions for all metabolites analyzed by GC-MS in *Figure 3*.
DOI: https://doi.org/10.7554/eLife.27713.019

**Figure supplement 1.** Identification of cystine as the metabolite in standard culture media that potentiates the glutaminase inhibitor CB-839.
DOI: https://doi.org/10.7554/eLife.27713.017

**Figure supplement 2.** Glutathione depletion and ROS are not solely responsible for decreased cell proliferation following glutaminase inhibition in high cystine environments.
DOI: https://doi.org/10.7554/eLife.27713.018

alone at RPMI levels to adult bovine serum increased glutamine contribution to the TCA cycle to the same extent as addition of the whole pool of all RPMI nutrients (*Figure 3A*). Addition of amino acids alone at RPMI levels also induced CB-839 sensitivity (*Figure 3B*), as did addition of amino acids to DMEM levels (*Figure 3—figure supplement 1A*). We next sought to identify the amino acid(s) causing this increased use of glutamine. Because DMEM contains fewer amino acids, and causes the same phenotype as RPMI amino acids, we reasoned that the amino acids responsible must be present in DMEM. Therefore, we systematically determined CB-839 sensitivity of cells cultured in adult bovine serum supplemented with these amino acids minus subsets that share common transport mechanisms (*Hediger et al., 2013*; *Hyde et al., 2003*). Adding the DMEM amino acid pool lacking serine, glycine, threonine and cystine to adult bovine serum failed to cause increased sensitivity of cells to CB-839, while omitting other subsets of amino acids had no effect on CB-839 sensitivity (*Figure 3—figure supplement 1A*). These data argue that one or more of these four amino acids is responsible for the phenotype. Serine, glycine, threonine or cystine were individually increased to DMEM levels in adult bovine serum containing DMEM levels of other amino acids. Only cystine addition triggered sensitivity to CB-839 (*Figure 3—figure supplement 1B*). Addition of cystine alone to adult bovine serum is sufficient to sensitize cells to CB-839 (*Figure 3B*). Similarly, addition of cystine at RPMI levels to adult bovine serum causes an increase in glutamine labeling of TCA cycle intermediates that is comparable to the labeling pattern observed in cells cultured in adult bovine serum containing levels of all nutrients found in RPMI (*Figure 3A*). Cystine levels in adult plasma (~20–50 µM) are 4–10 fold lower than in DMEM or RPMI (~200 µM) (*Table 1*). These experiments demonstrate that high levels of cystine found in standard tissue culture formulations cause enhanced glutamine anaplerosis and dependence.

Glutamine derived glutamate has multiple possible fates in cells (*Figure 3—figure supplement 2A*). It can contribute to TCA cycle metabolites, but also contributes to other metabolites like glutathione. Previous studies have suggested that glutamine-derived glutathione is essential for cell growth and further that depletion of glutathione upon glutaminase inhibition and subsequent loss of reactive oxygen species (ROS) buffering forms the basis of glutaminase dependence (*Biancur et al., 2017*; *Jin et al., 2015*; *Li et al., 2015*; *Timmerman et al., 2013*; *Ulanet et al., 2014*). In contrast, other studies have found that maintenance of TCA cycle derived metabolite levels, such as nucleotides, forms the basis of glutaminase dependence (*Cox et al., 2016*; *Gaglio et al., 2009*; *Lee et al., 2016*; *Okazaki et al., 2017*; *Patel et al., 2016*). We examined whether cystine-induced glutaminase dependence in A549 cells could be specifically ascribed to TCA cycle or glutathione depletion. Both TCA cycle metabolites and reduced glutathione (GSH) levels were decreased in A549 cells cultured in RPMI and exposed to CB-839 (*Figure 3—figure supplement 2B*), suggesting that high cystine-driven use of glutaminase is required for both TCA cycle anaplerosis and maintenance of glutathione. CB-839 treatment also caused increased levels of DCFDA staining, indicating ROS levels increase following glutaminase inhibition, which was reversible by co-treatment with N-acetylcysteine (NAC) (*Figure 3—figure supplement 2C*). These data show that both glutathione depletion and TCA cycle depletion occur upon glutaminase inhibition in the presence of high extracellular cystine.

We next performed rescue experiments with dimethyl α-ketoglutarate or glutathione monoethyl ester to determine if replenishment of the TCA cycle or glutathione could prevent the anti-proliferative effect of glutaminase inhibition in RPMI which contains high levels of cystine. We also assessed if glutathione depletion with buthionine sulfoximine, an inhibitor of GSH synthesis, had similar anti-proliferative effects as glutaminase inhibition with CB-839, and if NAC treatment could reverse the effects of CB-839 treatment. Treatment of cells with dimethyl α-ketoglutarate or glutathione monoethyl ester was able to rescue the proliferation of CB-839 treated A549 cells, albeit to differing extents (*Figure 3—figure supplement 2D*). However, assessment of TCA cycle metabolite and GSH levels in cells rescued with dimethyl α-ketoglutarate, showed that this rescued both TCA metabolite pools and GSH levels (*Figure 3—figure supplement 2E*). Similarly, glutathione monoethyl ester treatment increased levels of both TCA cycle metabolites and GSH in CB-839 treated cells, although glutathione monoethyl ester was less capable of rescuing both the anti-proliferative effect of CB-839 and metabolite levels at the concentration used (*Figure 3—figure supplement 2E*). These data are consistent with previous reports showing A549 cells express γ-glutamyl transferase and are able to degrade GSH to replenish glutamate pools (*Kang et al., 1994*). Thus, neither α-ketoglutarate or GSH rescue provided definitive insight into whether TCA cycle intermediates or GSH levels are more limiting for proliferation in high cystine upon glutaminase inhibition. However, as described

previously (*Kang et al., 1991*), we found that BSO treatment had no effect on A549 proliferation (*Figure 3—figure supplement 2D*) despite GSH depletion to undetectable levels (*Figure 3—figure supplement 2E*). NAC treatment, while able to decrease DCFDA detectable ROS (*Figure 3—figure supplement 2C*), did not rescue cell proliferation following CB-839 treatment (*Figure 3—figure supplement 2D*). Thus, while we have not identified precisely which glutamate dependent metabolic processes are limiting for proliferation upon glutaminase inhibition in high cystine conditions, these data argue that glutathione depletion and the subsequent increase in ROS alone is not sufficient to explain decreased proliferation of these cells.

## Cystine driven glutamine anaplerosis requires xCT/SLC7A11 expression

How might exogenous cystine regulate the contribution of glutamine carbon to the TCA cycle? Glutamine metabolism is linked to cystine metabolism via the $x_c^-$ transporter system. This transporter system is composed of the transporter xCT (encoded by *SLC7A11*) and the chaperone 4F2hc/CD98 (encoded by *SLC3A2*), and together they mediate the exchange of glutamate for cystine across the plasma membrane (*Lewerenz et al., 2013*). We hypothesized that in the presence of high exogenous cystine, xCT-mediated transport of glutamate might deplete the intracellular glutamate/αKG pool, thus promoting glutamate regeneration from glutamine via glutaminase (*Figure 4A*). To begin to test this, we measured glutamate release and uptake of both glutamine and cystine by A549 cells cultured in RPMI or RPMI containing low levels (10 μM) of cystine. Consistent with the hypothesis that cystine potentiates glutamine anaplerosis by triggering enhanced glutamate secretion via xCT and increased glutamine catabolism, we observed that higher extracellular cystine increased both the glutamine and cystine uptake rates as well as the glutamate release rate (*Figure 4—figure supplement 1*) in A549 cells. To determine if xCT/*SLC7A11* was required for cystine induced glutamine anaplerosis and dependence, we generated A549 cells with shRNA-mediated stable knockdown of *SLC7A11*, where knockdown was or was not rescued by expression of an shRNA-resistant *SLC7A11* cDNA (*Figure 4B*). We first confirmed that knockdown of *SLC7A11* with two hairpins reduced the glutamine uptake and glutamate release rate of cells cultured in RPMI, but did not have a detectable effect on glutamine uptake and glutamate release when cells were cultured in RPMI containing low cystine (10 μM) (*Figure 4—figure supplement 2A*). Knockdown of *SLC7A11* had no consistent effect on steady state intracellular levels of glutamate or the glutamate derived metabolite glutathione in either RPMI or RPMI with low cystine (*Figure 4—figure supplement 2B*). We next traced the fate of [U-$^{13}$C$_5$]glutamine in *SLC7A11* knockdown or control cells when cultured in adult bovine serum or adult bovine serum supplemented with RPMI levels of cystine. To quantify the extent to which cystine enhanced glutamine anaplerosis, we assessed the difference between normalized m + 5 labeled αKG in adult bovine serum and in adult bovine serum with high cystine. We found that knockdown of *SLC7A11* substantially decreased the ability of cystine to potentiate glutamine anaplerosis, and that this effect was blunted by expression of the shRNA-resistant *SLC7A11* cDNA (*Figure 4C*). Additionally, knockdown of *SLC7A11* with multiple hairpins (*Figure 4D*) abrogated CB-839 sensitivity of A549 cells in adult bovine serum with RPMI levels of cystine (*Figure 4E*). Thus, xCT/*SLC7A11* is necessary for cystine induced glutamine anaplerosis and dependence in A549 cells.

We next asked if xCT/*SLC7A11* expression influences cystine-induced glutamine anaplerosis across other human cancer cell lines. We traced [U-$^{13}$C$_5$]glutamine fate in a panel of human cell lines from cancers arising in multiple tissues and with multiple genetic drivers (*Figure 4—source data 1*). These cells were cultured in RPMI or RPMI with low (10 μM) cystine and the extent of glutamine anaplerosis determined as in *Figure 4C*. We found a strong correlation between cystine-induced glutamine anaplerosis and xCT/*SLC7A11* mRNA expression reported by the Cancer Cell Line Encyclopedia (CCLE) (*Barretina et al., 2012*), suggesting that cystine levels and xCT/*SLC7A11* expression contribute to the prominent use of glutamine as a anaplerotic TCA cycle substrate in many cancer cells in culture (*Figure 4F*).

Given that xCT/*SLC7A11* expression correlates with the ability of cystine to induce glutamine anaplerosis, we reasoned that xCT/*SLC7A11* expression might also be a determinant of glutaminase dependence in cancer cells. We compared reported CB-839 IC50 values (*Gross et al., 2014*) and xCT/*SLC7A11* mRNA expression levels from the CCLE for a panel of breast cancer cell lines (*Figure 4G* and *Figure 4—source data 1*). Sensitive lines tend to have higher xCT/*SLC7A11* expression than resistant lines, suggesting that xCT/*SLC7A11* expression may contribute to a requirement for glutaminase in cultured cell lines.

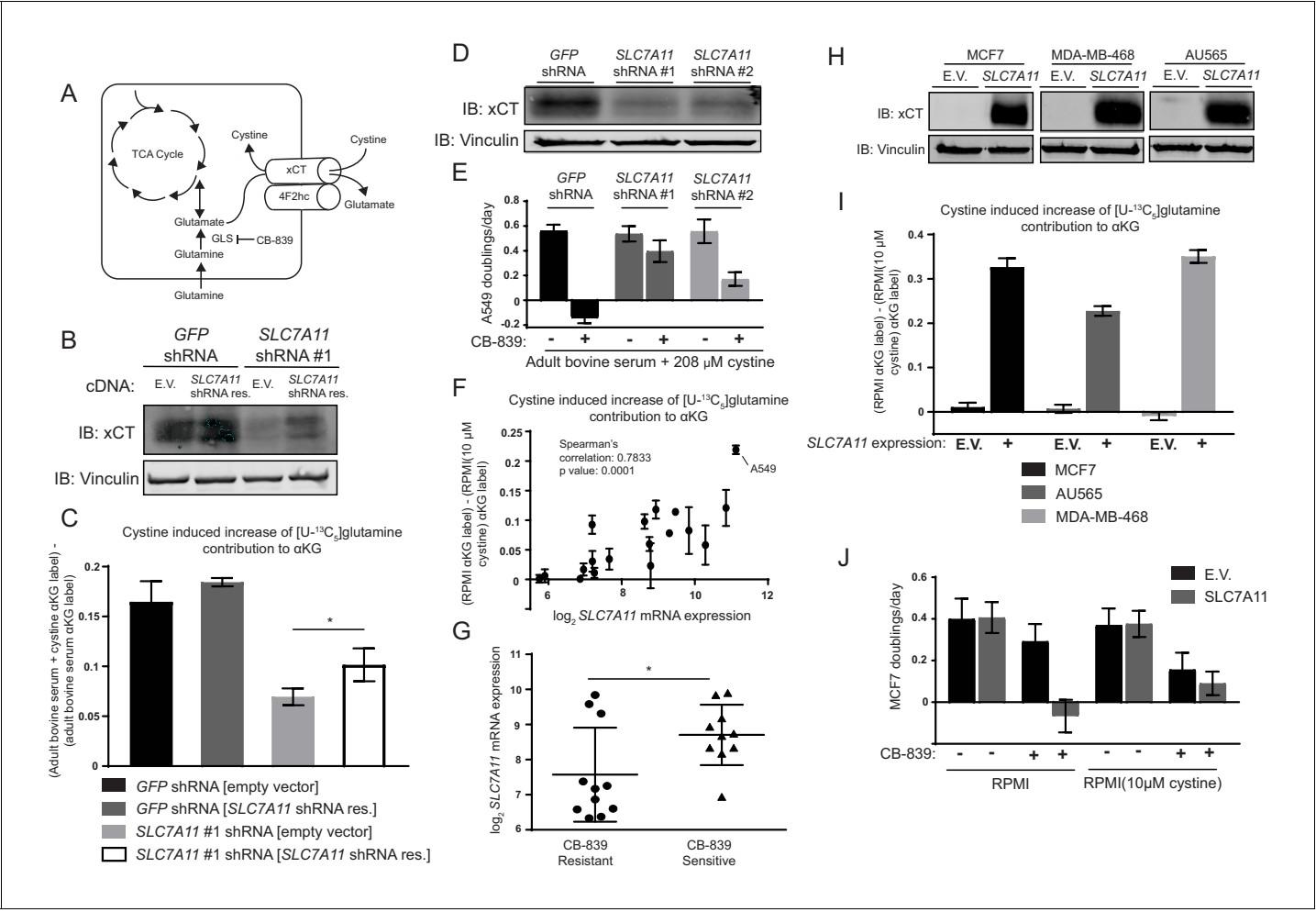

**Figure 4.** The cystine/glutamate antiporter xCT/*SLC7A11* is necessary and sufficient for cystine induced glutamine anaplerosis and CB-839 sensitivity. (A) System $x_c^-$ is a plasma membrane antiporter composed of two polypeptides, xCT (encoded by *SLC7A11*) and 4F2hc/CD98 (encoded by *SLC3A2*), that exchanges intracellular glutamate for extracellular cystine. (B) A549 cells were infected with lentiviruses encoding a *SLC7A11* targeting shRNA or a control shRNA targeting GFP as indicated. These cells were then infected with retroviruses expressing either shRNA resistant *SLC7A11* cDNA or empty vector (E.V.) as indicated. Shown is an immunoblot analysis of these modified cell lines for xCT protein expression with vinculin as a loading control. (C) The four cell lines from (B) were cultured for 8 hr in adult bovine serum or adult bovine serum with 208 µM cystine. Each medium included [U-$^{13}C_5$] glutamine added to ~33% enrichment (n = 3). M + 5 fractional labeling of α-ketoglutarate and glutamine for each cell line in each condition was determined. Shown is the difference in m + 5 fractional label of α-ketoglutarate (normalized to m + 5 fractional enrichment of glutamine) between adult bovine serum with 208 µM cystine and adult bovine serum. We define this as the 'cystine induced increase of glutamine contribution to αKG'. (D) Immunoblot analysis of A549 cells infected with lentiviruses encoding *SLC7A11* targeting shRNAs or a control shRNA targeting GFP as indicated. (E) Proliferation rates of cell lines from (D) cultured in adult bovine serum with 208 µM cystine with vehicle (DMSO) or 1 µM CB-839 is shown (n = 3). (F) Multiple cell lines (see *Figure 4—source data 1* for identity of cell lines) were cultured for 8 hr in RPMI or RPMI with 10 µM cystine. Each medium included [U-$^{13}C_5$]glutamine added to ~33% enrichment (n = 2–3). M + 5 fractional labeling of α-ketoglutarate and glutamine for each cell line in each condition was determined. Shown is 'cystine induced increase of glutamine contribution to αKG' defined as the difference of m + 5 fractional label of α-ketoglutarate (normalized to m + 5 fractional enrichment of glutamine) between RPMI and RPMI with 10 µM cystine for a given cell line. This term is plotted against *SLC7A11* mRNA expression data obtained from the cancer cell line encyclopedia (CCLE) (*Barretina et al., 2012*). (G) *SLC7A11* mRNA expression data from the CCLE is shown for breast cancer cell lines reported to be CB-839 resistant (IC50 >1 µM) or CB-839 sensitive (IC50 <1 µM) (*Gross et al., 2014*). Difference in *SLC7A11* expression between the two groups was tested by two-tailed unpaired t-test. (H) Indicated cell lines were infected with lentiviruses encoding *SLC7A11* cDNA or empty vector (E.V.). Shown is an immunoblot analysis of xCT protein expression for these modified cell lines with vinculin expression presented as a loading control. (I) Cystine induced increase of glutamine contribution to αKG was determined as in (F) for the cell lines described in (H). (J) Proliferation rates for MCF7 cells without (E.V.) or with SLC7A11 expression cultured in RPMI or RPMI with 10 µM cystine in the presence of vehicle (DMSO) or 1 µM CB-839 as indicated. For all panels, values represent the mean and the error bars represent ± SD. P values were calculated using a two-tailed unpaired t tests. * indicates p<0.05; ** indicates p<0.01; *** indicates p<0.001.
DOI: https://doi.org/10.7554/eLife.27713.020

The following source data and figure supplements are available for figure 4:

*Figure 4 continued on next page*

*Figure 4 continued*

**Source data 1.** Mass isotopomer distributions for all metabolites analyzed by GC-MS in *Figure 4*.
DOI: https://doi.org/10.7554/eLife.27713.025
**Figure supplement 1.** Extracellular cystine alters both cellular glutamine and cystine consumption and glutamate release rates.
DOI: https://doi.org/10.7554/eLife.27713.021
**Figure supplement 2.** Knockdown of *SLC7A11* reduces cellular glutamine uptake and glutamate release potentiated by high environmental cystine.
DOI: https://doi.org/10.7554/eLife.27713.022
**Figure supplement 3.** *SLC7A11* expression enhances cellular glutamine uptake and glutamate release potentiated by high environmental cystine.
DOI: https://doi.org/10.7554/eLife.27713.023
**Figure supplement 4.** Overexpression of xCT/*SLC7A11* causes cystine-induced CB-839 sensitivity for MDA-MB-468 and AU565 breast cancer cell lines.
DOI: https://doi.org/10.7554/eLife.27713.024

We asked if increased xCT/*SLC7A11* expression in CB-839 resistant cell lines would be sufficient to cause cystine-induced glutamine anaplerosis and increase glutaminase inhibitor sensitivity. xCT/*SLC7A11* was expressed in three CB-839 resistant breast cancer cell lines, all of which had low baseline expression of this transport system: MCF7, MDA-MB-468 and AU565 (*Figure 4H*). We first confirmed that expression of *SLC7A11* in MCF7 cells increased the glutamine uptake and glutamate release rate of cells cultured in RPMI, and that this effect was enhanced by high cystine levels in the media (*Figure 4—figure supplement 3A*). *SLC7A11* expression had no effect on intracellular glutamate or GSH levels in standard RPMI, but did prevent the decrease in glutathione levels observed when MCF7 cells are cultured in RPMI with low cystine (*Figure 4—figure supplement 3B*). We next traced [U-$^{13}$C$_5$]glutamine fate in these cell lines with and without *SLC7A11* overexpression. xCT/*SLC7A11* expression increased the incorporation of glutamine carbon into TCA cycle intermediates in the presence of RPMI levels of cystine in all three cell lines (*Figure 4I*). xCT/*SLC7A11* expression also potentiated CB-839 sensitivity when these cells are cultured in RPMI, but not RPMI with 10 µM cystine (*Figure 4J* and *Figure 4—figure supplement 4*). These results demonstrate that xCT/*SLC7A11* is necessary and sufficient for increased glutamine anaplerosis and glutaminase addiction in the presence of high levels of extracellular cystine.

## Low environmental cystine availability limits glutamine anaplerosis in vivo

Lastly, we asked whether the relatively low level of cystine available in vivo could limit glutamine anaplerosis of xCT/*SLC7A11*-expressing tumors. For these studies, we sought to increase cystine availability in A549 tumor bearing mice and monitor glutamine carbon incorporation into metabolites in the tumors. We found that an oral dose of 2.4 g/kg cystine could raise plasma cystine levels in mice from ~15 µM to ~70 µM for at least 4 hr (*Figure 5A*). We next administered cystine and [U-$^{13}$C$_5$]glutamine to A549 tumor bearing mice to determine if cystine administration would increase glutamine anaplerosis. Because oral cystine administration to mice surgically prepared for long-term glutamine infusion as performed in *Figure 1B* was not technically feasible, [U-$^{13}$C$_5$]glutamine was administered to tumor bearing mice via multiple bolus intravenous injections (*Elgogary et al., 2016*; *Lane et al., 2015*; *Tardito et al., 2015*; *Yuneva et al., 2012*) 30 min after cystine dosing (*Figure 5B*). This labeled glutamine delivery method does not allow us to assess steady state contribution of glutamine carbon to the TCA cycle in tumors; however, glutamine intravenous bolus injections do allow examination of glutamine incorporation into glutamate and downstream metabolites during a presteady state, kinetic labeling period and this method has previously been used to corroborate predicted changes in tumor glutamine utilization (*Elgogary et al., 2016*; *Lane et al., 2015*; *Tardito et al., 2015*; *Yuneva et al., 2012*) (*Figure 5B*). From measuring glutamine uptake and glutamate release rates under high and low cystine conditions (*Figure 4—figure supplement 1*), we surmise that the intracellular pools of glutamine and glutamate turnover more rapidly with increasing extracellular cystine. Thus, a prediction of our model that would be captured by kinetic [U-$^{13}$C$_5$]glutamine labeling is that intratumoral labeling of glutamine, as well as glutamate and TCA cycle metabolites, will be higher in the presence of exogenous cystine. Terminal [U-$^{13}$C$_5$]glutamine plasma enrichment was similar in cystine-treated and untreated animals, while cystine-treated animals had substantially higher plasma cystine concentrations (*Figure 5B*). Consistent with increased glutamine catabolism in cystine treated tumors, labeling of glutamine, glutamate and αKG from [U-$^{13}$C$_5$]

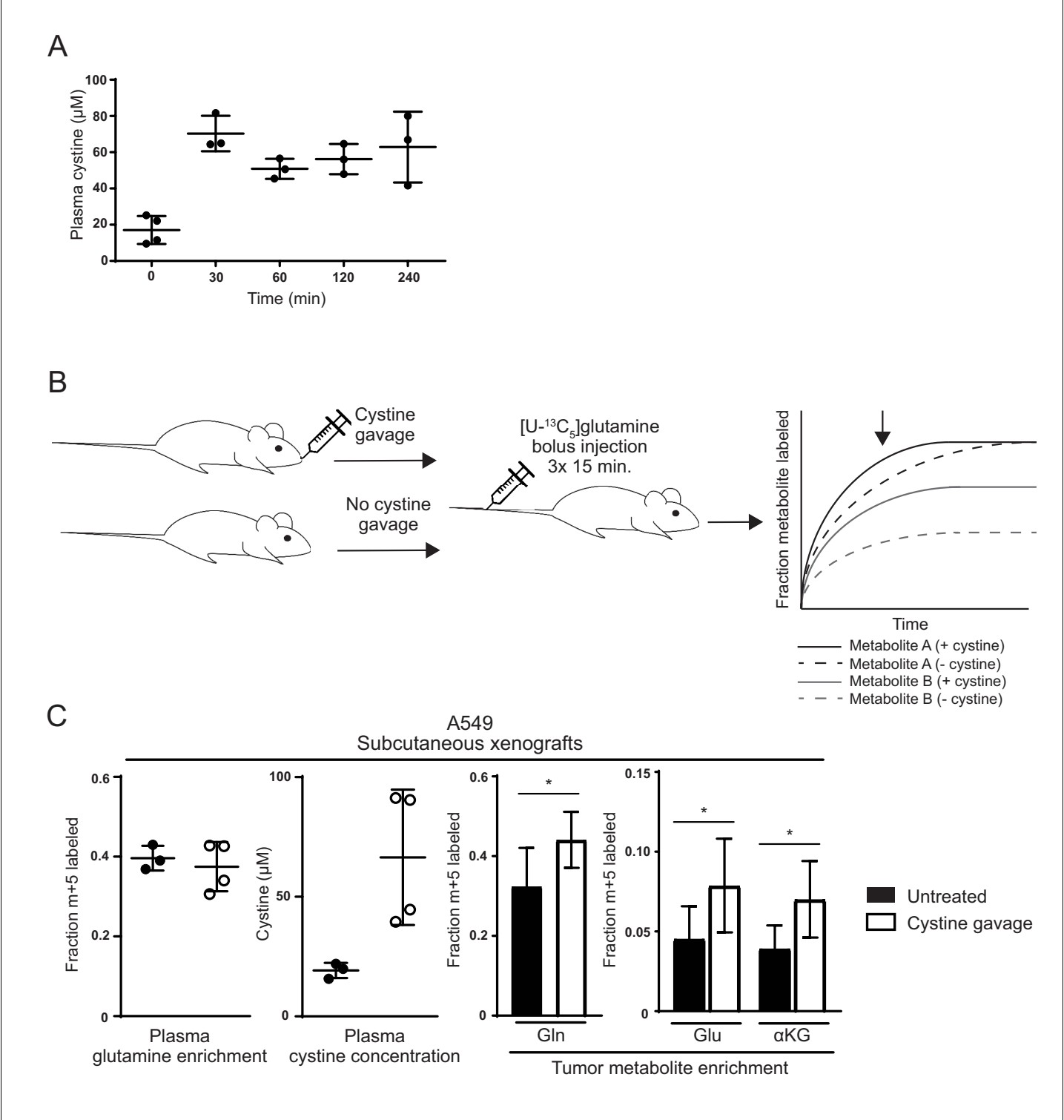

**Figure 5.** Raising cystine levels increases glutamine metabolism in tumors. (**A**) nu/nu mice were treated with 2.4 g/kg cystine by oral gavage, and plasma from these animals was collected at the indicated time points. Cystine concentration in plasma at each time point as determined by GC-MS is shown. (**B**) Schematic diagram of cystine administration to tumor bearing mice, followed by bolus I.V. injections of [U-$^{13}$C$_5$]glutamine, and analysis of intratumoral metabolite labeling during the period of kinetic labeling. Shown in the graph are hypothetical labeling patterns of two metabolites over time after glutamine injection. Metabolite A (shown in black) is labeled by glutamine faster in the presence of cystine, but the level of steady state labeling is unaffected. Metabolite B (shown in gray) is labeled by glutamine faster in the presence of cystine and also to a greater extent at steady state. We harvest and analyze tumors in the pre-steady state period of labeling as indicated by the arrow, and thus it is not possible to determine to

*Figure 5 continued on next page*

*Figure 5 continued*

what extent an increase in labeling is due to faster labeling kinetics or increased steady state contribution of the label to the given metabolite. (C) A549 tumor bearing mice were treated (n = 4) or not (n = 3) with oral cystine prior to three bolus [U-$^{13}$C$_5$]glutamine injections every 15 min. Plasma and tumor tissue was then harvested 15 min after the last injection (described in Materials and methods). *Left* Enrichment of m + 5 glutamine and cystine concentration in the plasma of these animals. *Right* Fractional labeling of glutamine (m + 5), glutamate (m + 5), α-ketoglutarate (m + 5) in A549 tumor tissue. For all panels, the values represent the mean and the error bars represent ± SD. P values were calculated using a two-tailed unpaired t tests. * indicates p<0.05; ** indicates p<0.01; *** indicates p<0.001.

DOI: https://doi.org/10.7554/eLife.27713.026

The following source data is available for figure 5:

**Source data 1.** Mass isotopomer distributions for all metabolites analyzed by GC-MS in *Figure 5*.

DOI: https://doi.org/10.7554/eLife.27713.027

glutamine in tumors was higher in cystine treated mice (*Figure 5C*). These results suggest that increasing cystine levels in tumors can cause increased glutamine consumption and glutamate release, a phenomenon we have shown using in vitro models leads to enhanced glutamine anaplerosis. These results are consistent with a model where low cystine levels in vivo limit tumor glutamine catabolism and anaplerosis.

## Discussion

### Coincidence of xCT/SLC7A11 expression and high environmental cystine contribute to increased glutamine metabolism

Environmental differences between in vitro culture conditions and tumors in vivo can lead to differential reliance on glutamine anaplerosis, causing some cells to be addicted to glutamine catabolism in vitro. We found that non-physiological cystine levels in tissue culture can explain many of the differences in glutamine metabolism between lung cancer cells in vitro and in tumors. The glutamate/cystine antiporter xCT/*SLC7A11* is necessary for the observed cystine-induced glutamine anaplerosis in A549 cells. Thus, high levels of extracellular cystine and the cellular capacity to secrete glutamate in exchange for cystine cooperate to increase glutamine anaplerosis in these cells.

Increased glutamate secretion in the presence of cystine increases intracellular glutamate turnover, which in turn will increase the kinetic rate of glutamine label incorporation into both glutamate and TCA cycle intermediates. However, there are multiple carbon sources that can contribute to the pool of TCA cycle metabolites, raising the question of why increased glutamate secretion also results in increased anaplerotic contribution of glutamine carbon relative to other nutrients. The enzymology of glutaminase suggests one potential explanation. GLS1 activity is strongly inhibited by glutamate (*Cassago et al., 2012*; *Curthoys and Watford, 1995*) suggesting that the drop in intracellular glutamate driven by high cystine could raise glutaminase activity and increase glutamine contribution to the TCA cycle relative to other pathways. Thus, it could be the action of non-physiological cystine concentrations constantly depleting the intracellular glutamate pool that causes glutamine anaplerosis to become dominant.

xCT/*SLC7A11* expression correlates with the ability of cystine to increase glutamine anaplerosis in a panel of cell lines derived from tumors of different oncogenotype and tissue of origin (*Figure 4F*). This argues that xCT/*SLC7A11* expression may be necessary for environmental cystine to enhance glutamine anaplerosis regardless of tumor type, and is not a phenomenon limited to NSCLC and A549 cells. Indeed, examination of panels of human breast cancer cell lines have found that xCT/*SLC7A11* promotes glutamate secretion, providing further evidence that xCT/*SLC7A11* can promote glutamine catabolism in cell types beyond NSCLC (*Briggs et al., 2016*; *Timmerman et al., 2013*). Collectively, these experiments suggest that glutaminase dependence in culture for many cancer cells derived from various tumor types will be strongly influenced by both environmental cystine and xCT/*SLC7A11* expression.

## Therapeutic implications of tissue cystine levels and their relationship to glutamine anaplerosis

These findings have clear implications for the clinical use of glutaminase inhibitors that are being evaluated in trials to treat a variety of tumor types (https://clinicaltrials.gov/ct2/show/NCT02071862). First, assessing xCT/*SLC7A11* expression may help identify patients likely to benefit from these drugs. While many cancer cells are considered to be glutamine addicted, not every cancer cell line requires glutamine for proliferation in vitro (*Cetinbas et al., 2016*; *Cheng et al., 2011*; *Gross et al., 2014*; *Timmerman et al., 2013*; *van den Heuvel et al., 2012*). Glutamine dependence has been linked to various oncogenes including *MYC* (*Gao et al., 2009*; *Wise et al., 2008*), *RAS* (*Son et al., 2013*), *IDH* (*Matre et al., 2014*; *Seltzer et al., 2010*) and hormone receptor status in breast cancers (*Gross et al., 2014*). However, the genetic and biochemical basis for glutamine dependence remains poorly understood. Thus, understanding the underlying factors that cause increased reliance on glutamine is essential for identifying patients that would likely benefit from clinical glutaminase inhibitors. Identification of high xCT/*SLC7A11* expression as a marker for glutaminase inhibitor sensitivity may be useful to identify patients that are more likely to benefit from glutaminase inhibitor therapy regardless of genotype. Functional examination of glutamate/cystine exchange by tumor cells using imaging techniques such as xCT-specific PET probes (*Baek et al., 2012*) may also be helpful in selecting tumors that are dependent on glutamine metabolism.

xCT/*SLC7A11* expression is known to be governed by the antioxidant response transcription factor NRF2 (*Sasaki et al., 2002*). Interest in modulating the antioxidant response pathway has led to the development of a number of NRF2-activating compounds, some of which are already in clinical use (*Davies et al., 2016*; *Gao et al., 2014*; *Magesh et al., 2012*). xCT/*SLC7A11* expression triggered by NRF2-activating drugs can sensitize tumors to glutaminase inhibitors (*Sayin et al., 2017*). In addition, beyond the use of xCT/*SLC7A11* expression as a marker for glutaminase inhibitor responsiveness, raising tumor cystine levels might also be a way to induce or enhance glutamine addiction. We found that raising tissue cystine levels may increase glutamine anaplerosis in tumors. Therefore, co-administration of cystine with glutaminase inhibitors may enhance response to glutaminase inhibitors. Oral cystine administration can be effective to raise plasma cystine levels in humans (*Morin et al., 1971*), arguing that this strategy could be applicable to treating patients.

## Tissue culture systems that reflect nutrient availability in vivo can be more representative models of tumor metabolism

Eagle's medium and RPMI were formulated to identify the minimal set of nutrients required for mammalian cancer cells and leukocytes to rapidly proliferate (*Eagle, 1955*; *Moore et al., 1967*), not to match physiological levels of nutrients available in vivo. Thus, these media lack certain nutrients available to tumors in vivo and also contain other nutrients in excess to what is found in blood. Several studies examining tumor metabolism in mice and humans have suggested that the way nutrients are used in tumors can be different from what is observed in cell culture. While increased glucose consumption and lactate secretion observed in most cells in culture is also observed in tumors (*Davidson et al., 2016*; *Hensley et al., 2016*; *Marin-Valencia et al., 2012*; *Mashimo et al., 2014*; *Sellers et al., 2015*; *Tardito et al., 2015*), glutamine oxidation and anaplerosis can differ between both lung and glial tumors and cell lines derived from these tumors in culture (*Davidson et al., 2016*; *Marin-Valencia et al., 2012*; *Tardito et al., 2015*). Thus, standard tissue culture can cause some aspects of central carbon metabolism to be misrepresented and the identification of media that better model the environmental conditions in tumors may facilitate better understanding of cancer metabolism.

Previous studies have suggested that altering tissue culture media to better reflect levels of amino acids found in human plasma can allow more accurate modeling of tissue metabolism (*Schug et al., 2015*; *Tardito et al., 2015*). Culturing glioblastoma cells in serum-like medium (SMEM) results in decreased glutamine utilization and a requirement to net produce glutamine from other carbon sources, similar to what is observed in glioblastoma in vivo (*Tardito et al., 2015*). The differences between SMEM and standard media formulations that elicited this difference is unknown, but the fact that culturing cells in media with more physiological cystine levels resulted in a change in glutamine metabolism is consistent with our findings.

In culture systems, the bulk of TCA cycle carbon can be accounted for from glucose and glutamine. However, the sources of anaplerotic TCA cycle carbon in tumors are not fully understood despite the increased contribution of glucose anaplerosis in some settings (*Davidson et al., 2016*; *Hensley et al., 2016*; *Sellers et al., 2015*). Adult bovine serum differs from standard media in levels of many nutrients. Thus, additional nutritional differences beyond amino acids may contribute to why glutamine metabolism of A549 cells in serum more closely mimics A549 tumors in vivo and suggests factors other than cystine levels might be involved. Supporting this hypothesis, addition of cystine to adult bovine serum increases A549 incorporation of glutamine carbon into the TCA cycle, but not to the same levels observed when the same cells are cultured in RPMI (*Figure 3A*). Additionally, A549 cells grown in adult bovine serum are also more resistant to CB-839 than A549 cells grown in RPMI with low levels of cystine (10 μM) (*Figure 3B*). These results suggest that alternative anaplerotic carbon source(s) are available in adult bovine serum that contribute to further glutaminase independence. Identification of what contributes to the TCA cycle in serum may yield new insight into the anaplerotic carbon sources used by cells in vivo.

Beyond the contribution of oncogenic mutations to reprogramming cellular metabolism, environment also plays a fundamental role in determining the metabolic program of cancer cells. We find that environmental cystine level is one variable that can alter how nutrients are used by cancer cells. Many studies aim to identify the set of genes required for cancer cell proliferation, with the goal of identifying novel therapeutic targets (*Hart et al., 2015*; *Shalem et al., 2014*; *Wang et al., 2017*). Based on our results, selection of media conditions will be essential to maximize the in vivo relevance of targets identified by screens to find metabolic gene vulnerabilities. Our results motivate further studies to characterize the nutritional content of the tumor microenvironment, and development of tissue culture systems to propagate and study cells under such conditions, as this may help translate metabolic dependencies of cancer cell lines into better cancer therapies.

## Materials and methods

### Cell lines and culture

All cell lines used in this study were directly obtained from ATCC (Manassas, VA) (A549: ATCC Cat# CRM-CCL-185, RRID:CVCL_0023; HCT116: ATCC Cat# CCL-247, RRID:CVCL_0291; 143B: ATCC Cat# CRL-8303, RRID:CVCL_2270; MDAMB231: ATCC Cat# HTB-26, RRID:CVCL_0062; PC3: ATCC Cat# CRL-1435, RRID:CVCL_0035; NCIH1299: ATCC Cat# CRL-5803, RRID:CVCL_0060; TT: ATCC Cat# CRL-1803, RRID:CVCL_1774; MDAMB468: ATCC Cat# HTB-132, RRID:CVCL_0419; ASPC1: ATCC Cat# CRL-1682, RRID:CVCL_0152; HS578T: ATCC Cat# HTB-126, RRID:CVCL_0332; NCIH226: ATCC Cat# CRL-5826, RRID:CVCL_1544; NCIH1395: ATCC Cat# CRL-5868, RRID:CVCL_1467; MIAPACA2: ATCC Cat# CRM-CRL-1420, RRID:CVCL_0428; AU565: ATCC Cat# CRL-2351, RRID:CVCL_1074; BT474: ATCC Cat# CRL-7913, RRID:CVCL_0179; MCF7: ATCC Cat# HTB-22, RRID:CVCL_0031) and DMSZ (Braunschweig, Germany) (CAL120: DSMZ Cat# ACC-459, RRID:CVCL_1104) or were gifts from other laboratories (PANC1: RRID:CVCL_0480; EVSAT: RRID:CVCL_1207). All cell lines not obtained directly from ATCC or DMSZ were STR tested to confirm their identity prior to use (University of Arizona Genetics Core, Tucson, AZ). All cell lines were regularly tested for mycoplasma contamination using the Mycoprobe mycoplasma detection kit (R and D Systems, Minneapolis, MN). All cells were cultured in a Heracell (Thermofisher, Waltham, MA) humidified incubators at 37°C and 5% $CO_2$. Cell lines were routinely maintained in RPMI-1640 or DMEM (Corning Life Sciences, Tewksbury, MA) supplemented with 10% heat inactivated fetal bovine serum (VWR Seradigm, Radnor, PA, Lot 120B14).

For continuous maintenance of A549 cells in adult bovine serum, the following modifications to standard tissue culture practices were made to maintain A549 cell viability during passage. First, cells were cultured in large volumes of adult bovine serum (8 mL/35 mm diameter well) to prevent depletion of essential nutrients during the culture. Additionally, the adult bovine serum was replaced with fresh adult bovine serum every 48 hr. Second, passaging serum maintained cells using standard trypsin (0.025%)/EDTA solution to detach cells leads to significant loss of viability. Therefore, serum grown cells were detached with a 1:1 mixture of 0.5% trypsin-EDTA (Thermofisher) and serum free Eagle's minimal essential media (Thermofisher). This allowed routine passaging and plating of cells with minimal loss of viability.

In unpublished observations made while culturing many cells lines in adult bovine serum, we have found that not all cell lines proliferate as standard monolayer cultures in adult bovine serum as they do in RPMI. Preliminarily, we have observed that some of these cells lines can proliferate in adult bovine serum when coated with collagen I according to previously published methods (*Olivares et al., 2017*), or by the addition of cystine to slightly higher levels (20–50 μM) than those found in adult bovine serum.

## Media preparation and analysis

All stable isotope tracing and proliferation rate experiments were performed in RPMI or DMEM containing 10% heat inactivated fetal bovine serum (Seradigm, Lot 120B14) that was repeatedly dialyzed against saline (150 mM NaCl) using 3.5 kDa cutoff membranes (Thermofisher) to remove all small molecule metabolites from the serum.

Adult bovine serum (Sigma Aldrich, St. Louis, MO, Lot 16A041), adult bovine heparinized plasma (Pel-freeze, Rogers, AR) and pooled adult human serum (Innovative Research, Novi, MI, Lot 20211) were thawed at 37°C for 2 hr. and heat inactivated for 30 min. at 56°C then filtered through a 0.22 μm filter prior to use or storage at −20°C.

To analyze absolute concentrations of amino acids, pyruvate and lactate in adult bovine serum, adult heparinized plasma and adult human serum, 10 μL of the serum or plasma was added to 10 μL of isotopically labeled internal standards of amino acids (Cambrige Isotope Laboratory, Andover, MA, MSK-A2-1.2, CLM-1822-H-PK, and CLM-8699-H-PK), pyruvate (Cambridge Isotope Laboratories, CLM-2440-PK) and lactate (Sigma Aldrich, 485926). These mixtures were then extracted in 600 μL of ice cold HPLC grade methanol, vortexed for 10 min, and centrifuged at 21 kg for 10 min. 450 μL of each extract was removed and dried under nitrogen gas and stored −80°C until further analysis by GC-MS. Glucose concentration of serum or plasma was measured using a YSI-2900D Biochemistry Analyzer (Yellow Springs Instruments, Yellow Springs, OH) according to the manufacturer's instructions.

To generate adult bovine serum containing added amino acids, stock amino acid mixtures were generated by adding individual amino acid powders (all obtained from Sigma Aldrich) at appropriate ratios in an electric blade coffee grinder (Hamilton Beach, Glen Allen, VA, 80365). The amino acids were then mixed using 10 pulses on the espresso setting. An appropriate amount of the amino acid mix was added to adult bovine serum, which was then sterilized using a 0.22 μm filter prior to use.

To prepare the dialyzed media 'Adult bovine serum → RPMI or DMEM', 210 mL of RPMI or DMEM was dialyzed twice overnight at 4°C against 4 L of adult bovine serum using 70 mL 3.5 kDa cutoff dialysis cassettes (Thermofisher). For 'RPMI or DMEM → adult bovine serum', 210 mL of adult bovine serum was dialyzed against RPMI or DMEM as above. These media were sterilized using a 0.22 μm filter prior to use.

For addition of metabolites to RPMI-1640, glutathione monoethyl ester (Santa Cruz, Dallas, TX), N-acetylcysteine (Sigma), dimethyl α-ketoglutarate (Sigma) were added at the indicated concentrations to RPMI-1640, which was then sterilized using a 0.22 μm filter prior to use.

## Generation of stable cDNA or shRNA expressing cell lines

Stable cDNA or shRNA expressing cell lines were generated by lentiviral or retroviral infection for 24 hr followed by selection in RPMI-1640 containing 1 μg/mL puromycin or 500 μg/mL hygromycin B. Mock infected cells were similarly selected, and selection was considered complete when no viable cells were detected in these mock infection controls. All virally manipulated cells were maintained under antibiotic selection at indicated concentrations until used in experimental assays.

## Vectors

For reduction of xCT/*SLC7A11* expression, lentiviral pLKO human *SLC7A11* shRNA vectors (*SLC7A11* shRNA #1: TRCN0000288926 and *SLC7A11* shRNA #2: TRCN0000288927) containing the puromycin resistance gene were obtained from Sigma Aldrich. A pLKO vector targeting GFP was used as a control.

To rescue xCT/*SLC7A11* expression, cDNA encoding human *SLC7A11* was obtained from Origene (Rockville, MD). The cDNA was then PCR amplified and cloned into the HindIII and ClaI sites of retroviral vector pLHCX, which contains the hygromycin resistance gene (Clontech, Mountain

View, CA), allowing for CMV promoter driven expression of *SLC7A11*. Site directed mutagenesis (*Hemsley et al., 1989*) was performed to synonymously mutate the sequence of *SLC7A11* targeted by *SLC7A11* shRNA #1: TRCN0000288926 (CCCTGGAGTTATGCAGCTAAT) such that it would no longer be targeted by this shRNA (GCCCGGCGTGATGCAATTGAT).

To overexpress *SLC7A11*, *SLC7A11* cDNA was PCR amplified and ligated into the SalI and NotI sites of the vector pENTR4(no ccDB) (Addgene, Cambridge, MA) (*Campeau et al., 2009*). LR Gateway recombination (Thermofisher) was used to clone the *SLC7A11* cDNA from pENTR4(no ccDB) into the expression vector pLenti CMV Puro DEST (w118-1) (Addgene) to allow for puromycin selectable CMV driven expression of *SLC7A11* (*Campeau et al., 2009*).

All vectors constructed for this study had the entire coding sequence confirmed by Sanger sequencing (Quintara Biosciences, Berkeley, CA) prior to use.

## Determination of cellular proliferation rates

Cellular proliferation rate in different media and drug conditions was determined as previously described (*Sullivan et al., 2015*). Briefly, cell lines proliferating in log phase in RPMI medium were trypsinized, counted and plated into six well dishes (Corning Life Sciences) in 2 mL of RPMI medium and incubated overnight. Initial seeding density was 20,000 cells/well for A549 cells, or 50,000 cells for MCF7, AU565 and MDA-MB-468 cells. The next day, a six well plate of cells was trypsinized and counted to provide a number of cells at the start of the experiment. Cells were then washed twice with 2 mL of phosphate buffered saline (PBS), and 8 mL of the indicated media premixed with indicated compounds or vehicles was added. This large volume of media was chosen to prevent severe nutrient depletion, especially when adding adult bovine serum medium. Cells were then trypsinized and counted 4 days after adding the indicated medias. Proliferation rate was determined using the following formula: Proliferation rate in doublings/day = [Log2(Final Day 4 cell count/Initial Day 0 cell count)]/4 days. Cells were counted using a Cellometer Auto T4 Plus Cell Counter (Nexcelom Bioscience, Lawrence, MA).

## Intracellular DCFDA staining

Measurement of intracellular DCFDA fluorescence was performed using the Abcam DCFDA Cellular Reactive Oxygen Species Detection Assay Kit (Abcam, Cambridge, MA, ab113851) according to the manufacturer's instructions. Briefly, 25,000 A549 cells were plated in each well of a 96 well plate in 200 µL RPMI without phenol red (Sigma Aldrich). Cells were allowed to attach for 8 hr. Subsequently, the media on the cells was changed to 200 µL fresh media containing DMSO, CB-839 and NAC as indicated. After 24 hr of treatment, the cells were washed twice with 200 µL of 1x Buffer included in the assay kit. Cells were then incubated with 100 µL of 1x Buffer containing 10 µM DCFDA for 45 min. at 37°C. An unstained control was included, and incubated with 100 µL of 1x Buffer alone. After staining, the DCFDA containing solution was removed and cells were resuspended 100 µL of PBS. DCFDA fluorescence was measured using an Infinite M200Pro plate reader (Tecan) in fluorescence mode with excitation at 485 and emission at 535. Cells were then detached with trypsin-EDTA solution and counted using a Cellometer Auto T4 Plus Cell Counter (Nexcelom Bioscience). The ratio of DCFDA signal intensity to cell number was subsequently computed.

## Preparation of cell extracts and immunoblotting

For immunoblotting analysis, cell lines growing in log phase were trypsinized, counted and plated at a density of 400,000 cells/well of a six well dish. The following day, cells were washed with 2 mL of PBS and then lysed in 100 µL RIPA buffer [25 mM Tris-Cl, 150 mM NaCl, 0.5% sodium deoxycholate, 1% Triton X-100, 1x cOmplete protease inhibitor (Roche)]. Cells were scraped and the resulting lysate was clarified by centrifugation at 21 kg for 20 min. Protein concentration of the lysate was determined by BCA assay (Thermofisher). Lysates were resuspended at 2 µg/µL in Laemmli SDS PAGE sample loading buffer (10% glycerol, 2% SDS, 60 mM Tris-Cl pH 6.8, 1% β-mercaptoethanol, 0.01% bromophenol blue) and denatured at 100°C for 5 min.

Extracts (30 µg of protein) were resolved by SDS PAGE using 10% acrylamide gels running at 120 V until the dye front left the gel. After SDS-PAGE resolution, protein extracts were transferred to nitrocellulose using an iBlot semidry transfer system (Thermofisher). Membranes were subsequently incubated with primary antibody in Odyssey buffer (Licor Biosciences, Omaha, NE), washed, and

incubated with IRDye-conjugated anti-mouse and anti-rabbit IgG secondary antibodies in Odyssey buffer with 0.1% Tween-20% and 0.02% SDS. Blots were imaged using an Odyssey infrared scanner (Licor Biosciences).

Antibodies and dilutions used in this study were: 1:1000 rabbit anti-xCT (Cell Signaling Technology, Danvers, MA, 12691S), 1:10000 mouse anti-Vinculin (Abcam, ab18058, RRID:AB_444215), 1:10000 IR680LT dye conjugated goat anti-rabbit IgG (Licor Biosciences, 925–68021), 1:10000 IR800 dye conjugated goat anti-mouse IgG (Licor Biosciences, 925–32210).

## Determination of cellular amino acid consumption and release rates

For amino acid consumption analysis, cell lines growing in log phase were trypsinized, counted and plated at a density of 100,000 cells/well of a six well dish and allowed to attach for 24 hr. The following day, cells were washed with 2 mL of PBS and then fed 2 mL of either RPMI or RPMI(10 μM cystine) media. 1 mL of media was immediately removed, spun for 3 min. at 845 g to remove any cells from the media, and then frozen at −80°C for later analysis. Cell number was also determined using a Cellometer Auto T4 Plus Cell Counter (Nexcelom Bioscience). 2 days later, media was harvested and cell number of the culture again determined. Amino acid concentration in the fresh or spent media was determined by GC-MS as described above in Media preparation and analysis. To calculate amino acid consumption rates, cell numbers at the initial and day two time points were used to fit an exponential growth function, and integration of these curves yielded the number of (cell · days) by which the media was conditioned. Changes in amino acid concentration for each culture were normalized to the integrated growth curve of each culture to yield amino acid consumption/release per cell per unit time.

## Cell culture isotopic labeling experiments and metabolite extraction

To determine steady state labeling of polar metabolites by glutamine in cultured cells, cell lines were seeded at an initial density of 200,000 cells/well in a six well dish in 2 mL of RPMI medium. Cells were incubated for 24 hr, and then washed twice with 2 mL of PBS. Cells were then incubated for 8 or 24 hr in the indicated media, to which [U-$^{13}$C$_5$]glutamine (Cambridge Isotope Laboratories, CLM-1822-H-PK) was added, such that the fractional enrichment of glutamine in the given medium would be ~33%.

Following the labeling period, media was aspirated from cells and the cells were rapidly washed in ~8 mL of ice cold saline. The saline was subsequently aspirated and 600 μL of ice cold methanol: water (4:1) was added. Cells were scraped on ice, and the resulting extracts were vortexed for 10 min, and centrifuged at 21 kg for 10 min. 450 μL of each extract was removed and dried under nitrogen gas and stored −80°C until further analysis.

## Liquid chromatography-mass spectrometry (LC-MS) analysis of intracellular metabolites

Cell cultures treated as indicated were washed in blood bank saline and extracted in methanol:water as described above in Cell culture isotopic labeling experiments and metabolite extraction. Cellular extracts were then resuspended in 100 μL of acetonitrile:water (1:1). LC-MS analysis was then performed using a QExactive orbitrap mass spectrometer using an Ion Max source and heated electrospray ionization (HESI) probe coupled to a Dionex Ultimate 3000 UPLC system (Thermofisher). External mass calibration was performed every 7 days.

Samples were separated by chromatography by injecting 10 μL of sample on a SeQuant ZIC-pHILIC 2.1 mm x 150 mm (5 μm particle size) column. Flow rate was set to 150 μL/min. and temperatures were set to 25°C for the column compartment and 4°C for the autosampler tray. Mobile phase A was 20 mM ammonium carbonat, 0.1% ammonium hydroxide. Mobile phase B was 100% acetonitrile. The chromatographic gradient was: 0–20 min.: linear gradient from 80% to 20% mobile phase B; 20–20.5 min.: linear gradient from 20% to 80% mobile phase B; 20.5 to 28 min.: hold at 80% mobile phase B.

The mass spectrometer was operated in full scan, polarity-switching mode and the spray voltage was set to 3.0 kV, the heated capillary held at 275°C, and the HESI probe was held at 350°C. The sheath gas flow rate was 40 units, the auxiliary gas flow was 15 units and the sweep gas flow was one unit. The MS data acquisition was performed in a range of 70–1000 m/z, with the resolution set

to 70,000, the AGC target at 1e5 and the maximum injection time at 20 msec. Relative quantitation of metabolites was performed with XCalibur QuanBrowser 2.2 (Thermofisher) using a five ppm mass tolerance, referencing an in-house retention time and m/z library of metabolite standards. In order to detect GSH and GSSG, the MS was operated in targeted selected ion monitoring (tSIM) mode with the quadrapole centered on M + H m/z 308.0811 (GSH) and m/z 613.1592, with an isolation window of 1 m/z. The resolution was set at 70,000, the AGC target was 1e5, and the maximum injection time was 250 ms.

## Animal studies

All experiments performed in this study were approved by the MIT Committee on Animal Care (IACUC). Nu/nu mice were purchased from Charles River (Wilmington, MA) (RRID:IMSR_CRL:088) and housed on a 12 hr light and 12 hr dark cycle, with ad lib access to food and water. For subcutaneous xenograft studies, mice were injected with 2,000,000 A549 cells (suspended in a volume of 100 μL PBS) per site into the right and left flank.

Continuous infusions were performed as previously described (*Davidson et al., 2016*). 3–4 days prior to tracer studies, tumor-bearing mice underwent a surgery to implant a catheter into the jugular vein and were allowed to recover. Mice were infused with [U-$^{13}$C$_5$]glutamine at 3.7 mg/kg/min for 300 min. At the end of the infusion, mice were terminally anesthetized with sodium pentobarbital and blood collected via heart puncture. Tissues were rapidly collected, freeze-clamped in liquid nitrogen, and stored −80°C.

Repeated bolus intravenous injection tracer studies were performed as previously described (*Lane et al., 2015*; *Yuneva et al., 2012*). Tumor-bearing mice were orally administered a 2.4 g/kg dose of L-cystine (Sigma Aldrich). Cystine was formulated as a 240 mg/mL suspension in 0.1% Tween 80% and 0.5% methylcellulose. Forty-five minutes after gavage, mice were lightly anesthetized using isofluorane and intravenously injected with 200 μL of [U-$^{13}$C$_5$]glutamine tracer (36.2 mg/ml dissolved in saline). These injections were performed a total of 3 times at 15 min intervals. Forty-five minutes after the first injection (and 15 min after the last injection), animals were euthanized, blood collected via heart puncture, and relevant tissues rapidly collected and freeze-clamped in liquid nitrogen.

## Tumor and plasma metabolite extraction

Frozen tissues were weighed (10–20 mg) and pulverized using a cryomill (Retsch, Haan, Germany). Metabolites were extracted in 1.3 mL chloroform:methanol:water (4:6:3), vortexed for 10 min, and centrifuged at 21 kg for 10 min. Polar metabolites were dried under nitrogen gas and stored −80°C until further analysis.

Blood collected from animals was immediately placed in EDTA-tubes (Sarstedt, North Rhine-Westphalia, Germany) and centrifuged to separate plasma. To analyze absolute concentrations of amino acids in the plasma, including cystine, 10 μL of plasma was added to 10 μL of a mixture of isotopically labeled amino acids of known concentrations (Cambridge Isotope Laboratories, MSK-A2-1.2). To analyze fractional enrichment of metabolites, 10 μL of plasma was diluted with 10 μL of water. Diluted plasma samples were extracted in 600 μL of ice cold HPLC grade methanol, vortexed for 10 min, and centrifuged at maximum speed for 10 min. 450 μL of each extract was removed and dried under nitrogen gas and stored −80°C until further analysis.

## Gas chromatograpy-mass spectrometry analysis of polar metabolites

Polar metabolites were analyzed by GC-MS as described previously (*Lewis et al., 2014*). Dried and frozen metabolite extracts were derivitized with 16 μL MOX reagent (Thermofisher) for 60 min. at 37°C. Samples were then derivitized with N-tertbutyldimethylsilyl-N-methyltrifluoroacetamide with 1% tert-butyldimethylchlorosilane (Sigma Aldrich) 30 min. at 60°C. Following derivitization, samples were analyzed by GC-MS, using a DB-35MS column (Agilent Technologies, Santa Clara, CA) installed in an Agilent 7890A gas chromatograph coupled to an Agilent 5997B mass spectrometer. Helium was used as the carrier gas at a flow rate of 1.2 mL/min. One microliter of sample was injected in split mode (all samples were split 1:1) at 270°C. After injection, the GC oven was held at 100°C for 1 min. and increased to 300°C at 3.5 °C/min. The oven was then ramped to 320°C at 20 °C/min. and held for 5 min. at this 320°C.

The MS system operated under electron impact ionization at 70 eV and the MS source and quadrupole were held at 230°C and 150°C respectively. The detector was used in scanning mode, and the scanned ion range was 100–650 m/z. Mass isotopomer distributions were determined by integrating appropriate ion fragments for each metabolite (*Lewis et al., 2014*) using in-house software (*Young et al., 2008*) that corrects for natural abundance using previously described methods (*Fernandez et al., 1996*).

## Acknowledgements

We thank all members of the Vander Heiden lab for many useful discussions and experimental advice. CB-839 was a generous gift of Craig Thomas (NIH National Center for Advancing Translational Sciences). Several cell lines used in this study were generously provided by Alec Kimmelman (New York University) and David Sabatini (Whitehead Institute for Biomedical Research). We also thank the many thousands of citizens who continue to protest policies of the federal government that damage scientific institutions and practice in the United States. Their continued service to the American scientific enterprise, which has made this work possible, is invaluable. This work was supported by grants to MGVH from the NIH (R01 CA168653, R01 CA201276, and P30CA1405141), the Lustgarten Foundation, SU2C, and the Ludwig Center at MIT. AM and LVD were supported by NIH Ruth Kirschstein Fellowships, F32CA213810 and F32CA210421 respectively. DYG received support from T32GM007753. MGVH is a Howard Hughes Medical Institute Faculty Scholar.

## Additional information

### Competing interests

Matthew G Vander Heiden: Is on the scientific advisory board of Agios Pharmaceuticals and Aeglea Biotherapeutics both of which seek to exploit altered metabolism for therapy. The other authors declare that no competing interests exist.

### Funding

| Funder | Grant reference number | Author |
| --- | --- | --- |
| National Institutes of Health | R01 CA168653 | Matthew G Vander Heiden |
| Lustgarten Foundation | Research Investigator Award | Matthew G Vander Heiden |
| Stand Up To Cancer | Innovative Research Grant | Matthew G Vander Heiden |
| Ludwig Institute for Cancer Research | Ludwig Center at MIT | Matthew G Vander Heiden |
| National Institutes of Health | R01 CA201276 | Matthew G Vander Heiden |
| National Institutes of Health | P30CA1405141 | Matthew G Vander Heiden |
| National Institutes of Health | F32CA213810 | Alexander Muir |
| National Institutes of Health | F32CA210421 | Laura V Danai |
| National Institutes of Health | T32GM007753 | Dan Y Gui |
| Howard Hughes Medical Institute | HHMI Faculty Scholar | Matthew G Vander Heiden |

The funders had no role in study design, data collection and interpretation, or the decision to submit the work for publication.

### Author contributions

Alexander Muir, Conceptualization, Formal analysis, Investigation, Methodology, Writing—original draft, Writing—review and editing; Laura V Danai, Conceptualization, Formal analysis, Investigation, Methodology, Writing—review and editing; Dan Y Gui, Conceptualization, Investigation, Methodology, Writing—review and editing; Chiara Y Waingarten, Formal analysis, Investigation, Writing—

review and editing; Caroline A Lewis, Resources, Formal analysis, Methodology, Writing—review and editing; Matthew G Vander Heiden, Conceptualization, Formal analysis, Supervision, Funding acquisition, Writing—review and editing

### Author ORCIDs
Alexander Muir http://orcid.org/0000-0003-3811-3054
Matthew G Vander Heiden http://orcid.org/0000-0002-6702-4192

### Ethics
Animal experimentation: All animals experiments were performed using protocols (#1115-110-18) that were approved by the MIT Committee on Animal Care (IACUC). All surgeries were performed using isoflurane anesthesia administered by vaporizer and every effort was made to minimize suffering.

### Decision letter and Author response
Decision letter https://doi.org/10.7554/eLife.27713.032
Author response https://doi.org/10.7554/eLife.27713.033

## Additional files

### Supplementary files
• Transparent reporting form
DOI: https://doi.org/10.7554/eLife.27713.028

### Major datasets
The following previously published dataset was used:

| Author(s) | Year | Dataset title | Dataset URL | Database, license, and accessibility information |
|---|---|---|---|---|
| Barretina J, Caponigro G, Stransky N, Venkatesan K, et al | 2012 | SNP and Expression data from the Cancer Cell Line Encyclopedia (CCLE) | https://www.ncbi.nlm.nih.gov/geo/query/acc.cgi?acc=GSE36139 | Publicly available at the NCBI Gene Expression Omnibus (accession no. GSE36139) |

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
