## [Decision Letter]

Thank you for submitting your article "Environmental cystine drives glutamine anaplerosis and sensitizes cells to glutaminase inhibition" for consideration by *eLife*. Your article has been reviewed by three peer reviewers, one of whom is a member of our Board of Reviewing Editors, and the evaluation has been overseen by Sean Morrison as the Senior Editor. The following individual involved in review of your submission has agreed to reveal his identity: Oliver Maddocks (Reviewer #3).

The reviewers have discussed the reviews with one another and the Reviewing Editor has drafted this decision to help you prepare a revised submission.

Summary:

These authors studied how glutamine dependence is regulated in culture and in vivo. They identify extracellular cystine as the key feature regulating glutamine-dependent anaplerosis and dependence on the enzyme glutaminase. These activities are enforced through expression of the xCT transporter, which imports cystine in exchange for glutamate. Silencing xCT reduces glutaminase dependence in culture and overexpressing xCT is sufficient to increase sensitivity to the glutaminase inhibitor CB839. Feeding tumor-bearing mice with high doses of cystine alters tumor handling of glutamine, indicating that extracellular nutrient availability can drive metabolic programs within tumors.

The reviewers were generally quite positive about the paper, but felt that it could be strengthened by addressing a few issues.

Essential revisions:

1) Most of the work focuses on a single cell line to establish the mechanism for glutamine dependence. The authors should provide some basic isotope tracing data (e.g. data shown in Figure 1) for some of the other cell lines introduced later in the paper. Perhaps the authors could simply derive these data from existing MS analysis such as the one shown in Figure 4.

2) Better characterizing the effect of manipulating glutamine/cystine metabolism would strengthen the paper. Can the authors report the effect of cystine concentration on rates of glutamine/cystine uptake and glutamate release? What is the impact of silencing/overexpressing SLC7A11 on intracellular glutamate/glutathione abundance and extracellular glutamate abundance? In conditions of cystine excess sensitizing cells to CB839, does the drug impact glutathione levels and ROS?

3) Rescue experiments to reverse the glutaminase-dependent phenotype would be useful. Does cell permeable α-ketoglutrate and/or GSH protect cells from CB839?

4) The in vivo analysis using 13C-glutamine is interesting, but the data do not entirely fit with the mechanism. Much (although not all, as the authors point out in Figure 5) of the gain in labeling of glutamate/AKG is transmitted from higher labeling in glutamine in the tumor. This indicates that part of the metabolic impact of cystine feeding is to increase glutamine uptake and/or to reduce de novo glutamine synthesis. Can the authors explain what is going on here?

5) While the central argument (increased cystine/glutamate exchange drives increased glutamine uptake to replenish intracellular glutamate) is logical, the specific link to anaplerosis seems not to be fully resolved. If glutamate availability is compromised by excessive secretion to import cystine, why does reversing this problem lead to enhanced entry of glutamine carbon into the TCA cycle as opposed to simple maintenance of a larger intracellular glutamate pool? Can this be explained in terms of changes in metabolic enzyme expression, or the need to fulfill some demand in the mitochondria?

---

## [Author Response]

*Essential revisions:*

*1) Most of the work focuses on a single cell line to establish the mechanism for glutamine dependence. The authors should provide some basic isotope tracing data (e.g. data shown in Figure 1) for some of the other cell lines introduced later in the paper. Perhaps the authors could simply derive these data from existing MS analysis such as the one shown in Figure 4.*

We agree with the reviewers and have provided data for additional cell lines. We performed glutamine tracing in RPMI and adult bovine serum (as in Figure 1) for six additional cell lines with distinct tissues of origin. The data from these experiments are now presented in Figure 1—figure supplement 3. Despite their diverse origins, all of the cell lines tested show reduced glutamine labeling of glutamate and TCA cycle intermediates when cultured in adult bovine serum compared to RPMI. These data show that reduced glutamine contribution to glutamate and TCA cycle intermediates occurs in multiple cell lines when cultured in adult bovine serum and is not limited to A549 cells.

*2) Better characterizing the effect of manipulating glutamine/cystine metabolism would strengthen the paper. Can the authors report the effect of cystine concentration on rates of glutamine/cystine uptake and glutamate release?*

We agree that determining how extracellular cystine levels affect glutamine uptake and glutamate release would provide additional support of our model that cystine uptake promotes glutamine catabolism via promoting glutamate export in an xCT-dependent manner. As suggested by the reviewers, we analyzed glutamine, glutamate and cystine consumption/release rates for A549 (xCT expressing) cells cultured in RPMI with high (208µM) or low (10µM) cystine. These data are presented in Figure 4—figure supplement 1. We find that cells cultured in high cystine consume both glutamine and cystine at higher rates, and release glutamate at a higher rate than cells cultured in low cystine. This is consistent with a model in which extracellular cystine promotes glutamate release, and this glutamate release promotes glutamine consumption to help maintain glutamate and glutamate-derived metabolites under high cystine conditions. We also note that Tardito et al. have previously found that high cystine levels increase glutamine consumption (Tardito et al., 2015) and that Timmerman et al. have previously found cystine consumption correlates with glutamate release (Timmerman et al., 2013).

*What is the impact of silencing/overexpressing SLC7A11 on intracellular glutamate/glutathione abundance and extracellular glutamate abundance?*

To address this question we measured glutamine uptake rates, glutamate release rates, intracellular glutamate levels, and intracellular glutathione levels in both A549 cells with *SLC7A11* knockdown (or control) and in MCF7 cells with increased *SLC7A11* expression (or control). These experiments were performed in RPMI with high (208µM) or low (10µM) cystine levels. These data are presented in Figure 4—figure supplement 2 and Figure 4—figure supplement 3. Consistent with previous reports, silencing or inhibiting *SLC7A11* decreases the release rate of glutamate (Briggs et al., 2016; Timmerman et al., 2013) and also decreases the glutamine consumption rate (Figure 4—figure supplement 2). We also observe that knockdown of *SLC7A11* only significantly affects glutamine consumption and glutamate release at high cystine levels (Figure 4—figure supplement 2). Similarly, we find that increasing xCT levels in MCF7 cells increases both glutamine consumption and glutamate release, and the extent to which *SLC7A11* expression increases these rates depends on cystine concentrations in the media (Figure 4—figure supplement 3). Despite these expected changes in glutamine uptake and glutamate release, there were no consistent changes in intracellular glutathione or glutamate levels upon *SLC7A11* knockdown. Culture in low cystine moderately decreased glutathione levels while increasing intracellular glutamate, regardless of *SLC7A11* expression status. *SLC7A11* overexpression in MCF7 cells did not lead to changes in GSH or glutamate levels in standard RPMI, but in low cystine RPMI, MCF7 cells overexpressing *SLC7A11* did have higher relative glutathione levels.

*In conditions of cystine excess sensitizing cells to CB839, does the drug impact glutathione levels and ROS?*

We thank the reviewers for bringing up this interesting point. There are numerous studies that suggest that glutamine catabolism is important for maintaining glutathione levels to buffer reactive oxygen species (ROS) (Biancur et al., 2017; Jin et al., 2015; Li et al., 2015; Timmerman et al., 2013; Ulanet et al., 2014). To address this question and determine if in this system glutamine derived glutamate contributed to glutathione and ROS buffering, we measured intracellular levels of TCA cycle metabolites and reduced glutathione (GSH) by LCMS in A549 cells grown in RPMI (a condition where cystine excess sensitizes cells to CB-839) with and without CB-839 treatment. We also measured dichlorofluorescin diacetate (DCFDA) fluorescence in A549 cells with and without CB-839 treatment and with n-acetylcysteine (NAC) treatment, as a proxy measurement for cellular ROS levels. We find that, in this high cystine condition, CB-839 not only depletes TCA cycle metabolites, but also depletes GSH and increases DCFDA signal. These results are presented in Figure 3—figure supplement 2. These data prompted us to perform further studies (described below) to determine if glutathione depletion was sufficient to prevent cellular proliferation, and if ROS scavenging by NAC would be able to reverse the anti-proliferative effect of CB-839 in high cystine cultures. In short, attempts to rescue the anti-proliferative effect of CB-839 in high environmental cystine by addition of TCA cycle intermediates or glutathione failed to show that depletion of either of these metabolites was specifically responsible for the toxicity of CB-389/high cystine. However, we did find that neither glutathione depletion nor accumulation of ROS that can be scavenged by NAC are sufficient to account for why CB-839 is toxic in the presence of high cystine.

*3) Rescue experiments to reverse the glutaminase-dependent phenotype would be useful. Does cell permeable α-ketoglutrate and/or GSH protect cells from CB839?*

We thank the reviewers for suggesting these experiments, as they are certainly of interest to help differentiate if TCA cycle anaplerosis or glutathione synthesis limits cell proliferation under conditions where cystine potentiates the effect of glutaminase inhibitors. We determined that both cell permeable α-ketoglutarate (aKG) and GSH could protect cells from glutaminase inhibition when cultured in RPMI (high cystine), albeit to different extents at the concentrations we tested. We performed metabolite analysis of treated cells, and found that aKG rescues both TCA cycle metabolite levels and GSH levels upon CB-839 treatment. We also found that GSH restores TCA cycle metabolite levels to the same extent that it rescues GSH levels in CB-839 treated cells. Thus, unfortunately, rescue with either aKG or GSH rescues both TCA cycle intermediates and GSH in cells and makes it impossible to differentiate if GSH depletion, TCA cycle intermediate depletion, or both explains the effect of glutaminase inhibitors on cell proliferation in high cystine.

To provide further insight into this important issue, we tested whether NAC could protect cells from CB-839, and if depletion of GSH with the GCL inhibitor BSO would be sufficient to prevent proliferation of A549 cells in RPMI. We found that NAC failed to rescue the proliferation of CB-839 treated A549 cells in RPMI (with high cystine). We also found that BSO treatment, while reducing glutathione levels below that seen with CB-839 treatment, had no effect on cell proliferation. Thus, while we cannot distinguish whether GSH or TCA cycle intermediate depletion limits proliferation following glutaminase inhibition in high cystine conditions, we can conclude that glutathione depletion and the subsequent increase in ROS are not sufficient to prevent proliferation.

All of the above results are included in Figure 3—figure supplement 2.

*4) The* in vivo *analysis using 13C-glutamine is interesting, but the data do not entirely fit with the mechanism. Much (although not all, as the authors point out in Figure 5) of the gain in labeling of glutamate/AKG is transmitted from higher labeling in glutamine in the tumor. This indicates that part of the metabolic impact of cystine feeding is to increase glutamine uptake and/or to reduce* de novo *glutamine synthesis. Can the authors explain what is going on here?*

We thank the reviewers for raising this issue, and apologize for not adequately explaining the experiment. For technical reasons, the in vivo tracing experiment in Figure 5 could not be performed using steady-state infusion as in Figure 1 (delivery of cystine by oral gavage leads to loss of the catheters needed for steady-state infusion of metabolites). Thus, we instead performed three I.V. bolus injections of (U-^13^C_5_)glutamine in the cystine treated or untreated mice as has been previously published (Elgogary et al., 2016; Lane et al., 2015; Tardito et al., 2015; Yuneva et al., 2012). Because bolus injections were used, the tracer will not reach steady state, and labeling has to be interpreted as a kinetic rather than a steady state labeling experiment. Therefore, the increase in glutamine labeling between the cystine treated and untreated mice do not necessarily represent a difference in the fraction of glutamine that tumors ultimately derive from extracellular glutamine versus other sources at steady state. Because this is a kinetic labeling experiment, the increased label may also be due to differences in the time that it takes cells to turn over and replace their glutamine pool in cystine treated versus untreated animals. We speculate that since cells exposed to extracellular cystine secrete glutamate and consume glutamine faster (Figure 4—figure supplement 1), they will turn over and replace their intracellular pool of glutamine with the supplied tracer faster than cells in the absence of high extracellular cystine. We note that TCA cycle labeling from labeled glutamine is higher in the cystine treated animals, but this experiment does not directly test whether increasing cystine levels in vivois sufficient to increase the steady state contribution of glutamine to the TCA cycle. Nevertheless, this experiment presents kinetic labeling patterns that are consistent with this hypothesis.

To ensure others do not misinterpret the experiment, we have revised the Results section discussing this experiment to: (1) explicitly state in the results and figure that this experiment is a kinetic rather than steady state experiment (we include a schematic in revised Figure 5 detailing how the experiment was done), (2) specifically discuss the possibility that the higher labeling of glutamine following cystine treatment is due to differences in glutamine turnover rate upon cystine administration, (3) make clear that this experiment does not explicitly show higher contribution of glutamine to the TCA cycle with cystine administration, but provides kinetic labeling that is consistent with this being the case, and (4) we have removed the data presented in original Figure 5, as we fear this data normalization may mislead readers to interpret this these data as if it were a steady state labeling experiment.

*5) While the central argument (increased cystine/glutamate exchange drives increased glutamine uptake to replenish intracellular glutamate) is logical, the specific link to anaplerosis seems not to be fully resolved. If glutamate availability is compromised by excessive secretion to import cystine, why does reversing this problem lead to enhanced entry of glutamine carbon into the TCA cycle as opposed to simple maintenance of a larger intracellular glutamate pool? Can this be explained in terms of changes in metabolic enzyme expression, or the need to fulfill some demand in the mitochondria?*

We think the reviewers are asking two questions here. The first question is whether cystine indeed forces glutamine to be catabolized anaplerotically. The second question is if glutamate availability is compromised by increased secretion to import cystine, why does reversing this problem lead to decreased (we believe the reviewers meant to say decreased here as we show in Figure 3–Figure 5 that lower cystine levels lead to decreased glutamine labeling of the TCA cycle) entry of glutamine carbon into the TCA cycle as opposed to simple maintenance of a larger intracellular glutamate pool?

To address the first question, we have shown that glutaminase inhibition of A549 cells cultured in RPMI (where high cystine potentiates glutamate release (Figure 4—figure supplement 1)) does reduce levels of TCA cycle intermediates (Figure 3—figure supplement 2). There is also a higher level of TCA cycle labeling from glutamine in A549 cells cultured in RPMI versus serum or RPMI with low cystine (Figure 1, Figure 4). Collectively, we interpret these results to suggest that high environmental cystine increases the anaplerotic use of glutamine to maintain TCA cycle metabolite levels. We also note that both aKG and GSH rescue levels of TCA cycle intermediates when cells are exposed to glutaminase inhibitors in high cytine (see point 3 above).

Regarding the second question, “If glutamate availability is compromised by excessive secretion to import cystine, why does reversing this problem lead to decreased entry of glutamine carbon into the TCA cycle as opposed to simple maintenance of a larger intracellular glutamate pool?” We would like to clarify that we do not mean to imply that cystine levels affect TCA cycle pool sizes, the rate of glutamate entry into the TCA cycle, or the anaplerotic demand of cells and mitochondria. Instead, we interpret our steady state labeling experiments to suggest that under high cystine conditions, the TCA metabolites pools (of whatever size they may be and at whatever speed they are kinetically labeled) are derived to a greater extent from glutamine than under low environmental cystine levels. This leads us to question, instead, why high cystine-driven glutamate release specifically increases the contribution of glutamine to the glutamate and TCA cycle metabolite pools. We speculate this could be due to the enzymology of glutaminase and have included a section in the Discussion specifically addressing this point:

“Increased glutamate secretion in the presence of cystine increases intracellular glutamate turnover, which in turn will increase the kinetic rate of glutamine label incorporation into both glutamate and TCA cycle intermediates. However, there are multiple carbon sources that can contribute to the pool of TCA cycle metabolites, raising the question of why increased glutamate secretion also results in increased anaplerotic contribution of glutamine carbon relative to other nutrients. The enzymology of glutaminase suggests one potential explanation. GLS1 activity is strongly inhibited by glutamate (Cassago et al., 2012; Curthoys and Watford, 1995) suggesting that the drop in intracellular glutamate driven by high cystine could raise glutaminase activity and increase glutamine contribution to the TCA cycle relative to other pathways. Thus, it could be the action of non-physiological cystine concentrations constantly depleting the intracellular glutamate pool that causes glutamine anaplerosis to become dominant.”

We hope this discussion clarifies the interpretation of our steady state glutamine tracing experiments, and addresses the reviewers’ questions.